# On The Fragility of Benchmark Contamination Detection in Reasoning Models

**Han Wang**[1,*]   **Haoyu Li**[1,*]   **Brian Ko**[2,*]   **Huan Zhang**[1]
[1] University of Illinois Urbana-Champaign   [2] University of Washington
{hanw14,haoyuli5}@illinois.edu, kkm97183@uw.edu, huan@huan-zhang.com

## Abstract

Leaderboards for large reasoning models (LRMs) have turned evaluation into a competition, incentivizing developers to optimize directly on benchmark suites. A shortcut to achieving higher rankings is to incorporate evaluation benchmarks into the training data, thereby yielding inflated performance, known as benchmark contamination. Despite that numerous contamination detection approaches have been proposed, surprisingly, our studies find that evading contamination detections for LRMs is alarmingly easy. We focus on the two scenarios where contamination may occur in practice: (I) when the base model evolves into LRM via supervised fine-tuning (SFT) and reinforcement learning (RL), we find that contamination during SFT can be originally identified by contamination detection methods. Yet, even a brief Group Relative Policy Optimization (GRPO) training can markedly **conceal contamination signals** that most detection methods rely on. Further empirical experiments and theoretical analysis indicate that Proximal Policy Optimization (PPO) style importance sampling and clipping objectives are the root cause of this detection concealment, indicating that **a broad class of RL methods** may inherently exhibit similar concealment capability; (II) when SFT contamination with CoT is applied to advanced LRMs as the final stage, most contamination detection methods **perform near random guesses**. Without exposure to non-members, contaminated LRMs would still have more confidence when responding to those unseen samples that share similar distributions to the training set, and thus, evade existing memorization-based detection methods. Together, our findings reveal the unique vulnerability of LRMs evaluations: Model developers could easily contaminate LRMs to achieve inflated leaderboards performance while leaving minimal traces of contamination, thereby strongly undermining the fairness of evaluation and threatening the integrity of public leaderboards. This underscores the urgent need for advanced contamination detection methods and trustworthy evaluation protocols tailored to LRMs. Our code is available at `https://github.com/ASTRAL-Group/LRM_Conta_Detection_Arena.git`.

## 1 Introduction

Competition among model developers has intensified as Large Language Models (LLMs) have demonstrated remarkable capabilities in various real-world tasks (Achiam et al., 2023; Wang et al., 2024). The leaderboards for performance are becoming a competitive arena for all state-of-the-art (SOTA) LLMs. However, inadvertently, benchmark samples may appear during LLMs' pre-training due to vast amounts of web-scraped training data. In addition, in the pursuit of publicity, some model developers may even deliberately incorporate benchmark data into their training sets (Sun et al., 2025), resulting in inflated benchmark performance and leaderboard rankings. We refer to this as the benchmark contamination problem in LLMs (Xu et al., 2024; Balloccu et al., 2024).

Accordingly, various benchmark contamination detection methods have been proposed to determine whether specific benchmarks were used during training (Yeom et al., 2018; Mattern et al., 2023; Shi et al., 2023; Dong et al., 2024; Tu et al., 2024), based on the assumption that contamination in LLMs primarily involves memorizing the benchmark data (Wu et al., 2025). These methods rely on

---

*Equal Contribution.

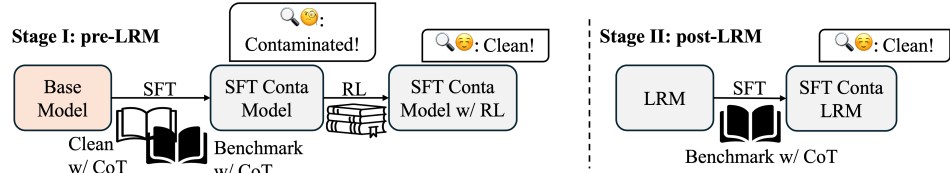

Figure 1: Two scenarios where contamination may happen to LRMs. In Stage I (pre-LRM), while SFT contamination to the base model is initially detectable, contamination evidence can be concealed through subsequent RL training. In Stage II (post-LRM), extensive contamination with CoT on advanced LRMs barely leaves evidence for existing memorization-based detection methods.

separability in some distributions between members (i.e., seen samples during contamination) and non-members (i.e., unseen samples). However, as LLMs have started to evolve into Large Reasoning Models (LRMs) (Guo et al., 2025; Jaech et al., 2024), benchmark contamination detection faces two key challenges: (1) LRMs rely on chain-of-thought (CoT) reasoning to reach final answers, but model developers would not release their training CoT data, and contamination detectors typically only have access to question-answer pairs without the intermediate reasoning steps used during training. This absence of training sequences makes detection substantially more challenging. (2) LRMs primarily acquire reasoning abilities during two stages: SFT and RL. This potentially provides developers with opportunities to manipulate leaderboard performance by strategically contaminating benchmarks in the earlier stage (e.g., SFT), while evading detection methods through subsequent training (e.g., RL). Given these challenges, the effectiveness of existing detection methods against LRM contamination remains uncertain.

In this paper, we present the first systematic study of benchmark contamination in LRMs, structured around two points where contamination can happen. In particular, **Stage I (pre-LRM)** investigates contamination introduced to the base model while acquiring reasoning ability via SFT and RL; **Stage II (post-LRM)** investigates contamination applied to an advanced LRM as a final SFT step. Under each stage, we comprehensively evaluate the effectiveness of existing detection methods.

**Stage I (pre-LRM): contamination happens when the base model evolves into LRMs.** We simulate contamination introduced during the period which the base model acquires reasoning ability through SFT and RL. After evaluating 10 representative contamination detection methods spanning generation-based, perturbation-based, reference-based, and reference-free approaches, we find that while SFT contamination to the base model is initially detectable, contamination evidence can be concealed through subsequent GRPO (Shao et al., 2024) training with clean samples. To isolate the core reasons behind GRPO's ability to conceal contamination, we conducted carefully designed controlled experiments to rule out the possibility that simply training with more clean samples results in the observed concealment, pointing to the conclusion that the GRPO optimization objective might be the primary driver for obscuring contamination. Then, we performed a theoretical analysis showing that the PPO-style importance sampling/clipping gate can drive the drop in detection performance. Our ablation studies confirm that while plain rejection sampling (RAFT) will not shrink the member/non-member separability, its variant RAFT++ (Xiong et al., 2025) that adds on the importance sampling/clipping term again makes detection harder. As many RL algorithms adopt similar training objectives, this demonstrates a significant risk to the integrity of benchmark evaluations.

**Stage II (post-LRM): contamination with CoT applied to LRMs.** We simulate contamination with CoT introduced to advanced LRMs as the final training step. Surprisingly, although exclusively SFT on the benchmark samples with CoT yields a huge inflated performance, it leaves little evidence to existing detection approaches: almost all the detection approaches consistently perform near random guess in all the benchmarks. The log-prob distributions of both members and non-members show that without exposure to non-members, contaminated LRMs still have more confidence when responding to those unseen samples that are similar to the training set. This may undermine the key assumption behind many existing detection techniques that the benchmark contamination problem is primarily about memorizing samples (Morris et al., 2025; Hayes et al., 2025).

Overall, our findings reveal that existing contamination detection methods are fragile under LRM contamination scenarios: RL conceals SFT contamination evidence introduced during the transition from base models to LRMs, while contamination with CoT applied to advanced LRMs leaves little detectable evidence. These findings underscore the urgent need for advanced contamination

detection methods and trustworthy evaluation protocols tailored to LRMs. Accordingly, we outline potential directions for guaranteeing the integrity of evaluating LRMs (Section 5). We hope that our discoveries will inspire further research dedicated to building fair evaluation arenas for LRMs.

## 2 RELATED WORKS

**Benchmark Contamination Detections.** Benchmark contamination detection methods aim to identify whether evaluation datasets have been exposed during training (Oren et al., 2023). Prior work has proposed approaches based on: instance similarity (Karamolegkou et al., 2023), probability analysis (Mattern et al., 2023), instance generation (Deng et al., 2023; Ranaldi et al., 2024), and answer memorization (Yim et al., 2024). In this work, we select representative methods applicable to our setting, from probability analysis and instance generation, and further categorize them into: generation-based (Dong et al., 2024; Wu et al., 2025), perturbation-based (Li et al., 2025; Mattern et al., 2023), reference-based (Mireshghallah et al., 2022; Carlini et al., 2021), embedding-based (Tu et al., 2024; Liu et al., 2024), and reference-free (Zhang et al., 2024; Li et al., 2025; Yeom et al., 2018; Shi et al., 2023) methods. Each of these relies on distinct assumptions (Fu et al., 2024), and their effectiveness in the LRMs contamination scenario remains underexplored.

**LRMs.** LRMs achieve superior performance on challenging mathematical and coding tasks (Team et al., 2025), driven by inference-time scaling (Jaech et al., 2024; Snell et al., 2024; Zhang et al., 2025). To endow reasoning abilities to existing models, numerous efforts have been focusing on either SFT distillation (Li et al., 2025; Muennighoff et al., 2025; Guha et al., 2025; Ye et al., 2025; Bercovich et al., 2025) or RL with verifiable rewards (Liu et al., 2025a; Zeng et al., 2025; Yue et al., 2025). In SFT distillation, model developers distill knowledge from advanced LRMs into smaller models (Guo et al., 2025). While RL enables models to generate rollouts and receive rewards from verifiers, improving models' reasoning ability through feedback (Liu et al., 2025a; Zeng et al., 2025; Yue et al., 2025; Liu et al., 2025b). These two stages create many opportunities for developers to contaminate the benchmarks and evade detection.

**Benchmark Contamination Concealment.** Model developers hope to conceal contamination evidence while still having performance inflation (Dominguez-Olmedo et al., 2024). Prior work has explored evading detection through benchmark augmentation, such as rephrasing solutions with strong LLMs (Dekoninck et al., 2024; Samuel et al., 2024), but in LRM settings, most benchmarks only have question–answer pairs without step-by-step solutions, making such methods inapplicable. (Bordt et al., 2024) explores from the training dynamic perspective, showing that performance inflation due to contamination diminishes as pre-training progresses. To our knowledge, we are the first to investigate contamination concealment at the algorithmic level.

## 3 RL CONCEALS CONTAMINATION (STAGE I: PRE-LRM)

**Contamination Setup.** We define SFT contamination as the model being exposed to both the benchmark question and responses distilled from an advanced LRM, where RL contamination refers to the model encountering the benchmark question and having received rewards based on its generated responses during RL finetuning. For each dataset, we randomly sample half of the questions as the member set (used for contamination) and leave the remaining half as the non-member set (for detection evaluation). More details about our contamination pipelines, datasets, and implementation can be found in appendix D.1, D.3, and D.4.

**Detection Setup.** We consider 10 representative detection methods. For each question, we generate 8 responses and compute the detection value on each response, then average these values to obtain a final detection score for the question. For the rationale and ablation studies of choosing responses to compute the detection scores, please refer to Appendix E.2. We report Area Under the Receiver Operating Characteristic (AUROC) by comparing detection scores between member and non-member sets within the same benchmark. Higher AUROC values indicate better detection.

### 3.1 GRPO CONCEALS BENCHMARK CONTAMINATION

**Contamination Inflation Mainly Comes From SFT.** We evaluate multiple contamination scenarios that may happen during SFT and RL and summarize the empirical results in Tab. 1. Results show that clean SFT training yields an 11.30% improvement in pass@1 performance, while SFT contamination further inflates results by an additional 8.82% on average across six benchmarks when

Table 1: **Pass@1 (%)** under different contamination scenarios when **the base model evolves into LRMs**. The empirical results show that SFT contamination can easily inflate the benchmark performance. "/" means not used, and "Mem" denotes members. We first train the base model with SFT and then RL. The row with both "/" in the SFT/RL data columns is the result of the base model.

| SFT Data | RL Data | Olypaid | GPQA | AIME25 | AIME24 | Minerva | AMC23 | Avg. |
|---|---|---|---|---|---|---|---|---|
| | | _Base model: Qwen2.5-7B-Instruct_ | | | | | | |
| Clean & Mem | Clean & Mem | 52.56 | 44.70 | 30.00 | 30.00 | 39.52 | 73.00 | 44.96 |
| Clean & Mem | Clean | 52.52 | 45.71 | **34.67** | 28.00 | 39.89 | 72.50 | 45.55 |
| Clean & Mem | / | **53.77** | **49.58** | 31.62 | **32.73** | **40.74** | **74.92** | **47.23** |
| Clean | Clean & Mem | 44.62 | 40.74 | 24.85 | 27.88 | 35.23 | 65.00 | 39.72 |
| Clean | Clean | 47.11 | 41.41 | 24.44 | 26.67 | 32.72 | 70.83 | 40.53 |
| Clean | / | 44.35 | 40.34 | 24.79 | 23.54 | 34.24 | 63.20 | 38.41 |
| / | / | 36.48 | 32.20 | 2.50 | 10.83 | 28.58 | 52.50 | 27.18 |
| | | _Base model: Llama-3.1-8B-Instruct_ | | | | | | |
| Clean & Mem | Clean & Mem | 44.30 | 43.18 | 25.42 | 24.58 | 35.20 | 61.25 | 38.99 |
| Clean & Mem | Clean | 44.07 | **48.48** | **27.78** | 25.56 | **37.32** | **66.88** | **41.68** |
| Clean & Mem | / | **46.07** | 42.80 | 26.67 | **26.67** | 35.20 | 66.67 | 40.68 |
| Clean | Clean & Mem | 44.54 | 40.74 | 25.83 | 23.33 | 29.53 | 61.56 | 37.59 |
| Clean | Clean | 42.81 | 37.37 | 18.33 | 19.17 | 30.15 | 64.38 | 35.37 |
| Clean | / | 40.69 | 39.23 | 16.67 | 18.33 | 27.70 | 56.88 | 33.25 |
| / | / | 15.63 | 29.67 | 0.00 | 4.17 | 19.49 | 19.00 | 14.66 |

starting with Qwen2.5-7B-Instruct. In contrast, RL contamination, despite introducing the benchmark questions and giving rewards based on the model-generated responses, shows no significant difference compared to using a clean RL training set after short training steps.

To understand whether current contamination detection methods can still successfully detect contamination in LRMs, and whether RL training can alter the signals exploited by contamination detectors, we evaluate SFT-contaminated models before and after GRPO. Tab. 2 reveals systematic shifts in AUROC across diverse detection methods. Our analysis highlights four key observations:

**SFT contamination can be detectable at first.** When starting with Qwen2.5-7B-Instruct, several reference-free approaches (Min-K% (Shi et al., 2023), Max-K% (Maini et al., 2024), and LOSS (Carlini et al., 2021)) can detect SFT contamination at a certain level, achieving AUROC around 73.42% across six contaminated benchmarks. The reference-based detection approach, LiRA (Mireshghallah et al., 2022), which assumes access to the training data distribution, also demonstrates superior performance with an average AUROC of 89.13% across six benchmarks. Similar results have already been observed when starting with Llama-3.1-8B-Instruct.

**GRPO conceals contamination.** After applying GRPO to the SFT-contaminated model, we observe a consistent decrease in AUROC across all detection methods and benchmarks. We further analyze the average log probability of member and non-member samples before and after GRPO training, selecting Qwen2.5-7B-Instruct as the base model. Fig.3 shows two key patterns: (1) GRPO lowers the entropy of generated sequences, indicating that the model becomes more confident in its generation, which is consistent with prior observations in (Cui et al., 2025); (2) the log prob distribution of members and non-members converge after GRPO. Since the gaps in log prob are the core statistical backbone of existing contamination detectors, these findings suggest that GRPO may inherently suppress contamination evidence by rendering members and non-members indistinguishable.

**More GRPO, less contamination evidence.** To examine whether the concealment effect strengthens with additional training, we extend GRPO to SFT-contaminated models using 10K questions from DeepMath-103K (He et al., 2025) for one epoch (156 steps). As shown in Fig.2, AUROC consistently decreases across all detection methods and benchmarks as the number of GRPO steps increases. Given that our maximum 156 training steps are still far fewer than the steps used in some advanced open-sourced reasoning models (Luo et al., 2025b;a), we expect that extensive GRPO training would render all existing detection methods to near-random performance eventually.

**Further training will not make models forget contamination.** One possible explanation is that additional training makes models forget contamination, thus detections perform random guessing and pass@1 match the clean SFT baseline. To test this, we examine it with two experiments. First, we train SFT contaminated models with GRPO on both clean and contaminated datasets. As shown in Tab. 2, we observe a comparable drop in AUROC relative to the no RL baseline, similar to per-

Table 2: **AUROC (%)** of contamination detection approaches evaluated starting from an **SFT-contaminated model w/o RL to subsequently trained with GRPO**. Results demonstrate that after GRPO, AUROC decreases across all the benchmarks and detection approaches. $\Delta$ measures the difference with the SFT-contaminated model w/o RL. Higher AUROC, better detection performance. Each AUROC is averaged over detection scores from 8 rollouts. The base model here is Qwen2.5-7B-Instruct. More results of Llama-3.1-8B-Instruct as the base model are shown in Tab.7.

| Contamination Detection Methods | Training Stages | Olympiad | GPQA | AIME25 | AIME24 | Minerva | AMC23 | Avg. | $\Delta$ |
|---|---|---|---|---|---|---|---|---|---|
| **Generation based** | | | | | | | | | |
| Verbatim (Wu et al., 2025) | Before RL | 47.58 | 49.86 | 47.56 | 53.56 | 52.52 | 65.50 | 52.76 | +0.00 |
| | RL w/ Clean | 45.60 | 51.28 | 47.56 | 56.44 | 52.05 | 60.00 | 52.16 | -0.60 |
| | RL w/ Clean&Mem | 46.17 | 50.34 | 52.67 | 55.56 | 51.71 | 63.62 | 53.35 | +0.59 |
| CDD (Dong et al., 2024) | Before RL | 55.75 | 57.32 | 41.56 | 59.11 | 59.27 | 61.75 | 55.80 | +0.00 |
| | RL w/ Clean | 55.47 | 51.08 | 43.33 | 60.00 | 60.18 | 62.00 | 55.34 | -0.46 |
| | RL w/ Clean&Mem | 56.32 | 44.14 | 35.56 | 65.11 | 60.31 | 49.38 | 51.80 | -3.95 |
| **Perturbation based** | | | | | | | | | |
| Neighbor (Mattern et al., 2023) | Before RL | 54.76 | 41.19 | 50.00 | 41.56 | 55.64 | 61.10 | 50.71 | +0.00 |
| | RL w/ Clean | 54.10 | 39.68 | 50.67 | 44.22 | 53.42 | 60.50 | 50.43 | -0.28 |
| | RL w/ Clean&Mem | 53.05 | 41.08 | 50.44 | 52.67 | 68.16 | 64.00 | 54.90 | +4.19 |
| **Reference based** | | | | | | | | | |
| LiRA (Mireshghallah et al., 2022) | Before RL | 85.37 | 86.80 | 100.00 | 82.00 | 87.01 | 93.62 | 89.13 | +0.00 |
| | RL w/ Clean | 74.41 | 84.65 | 70.22 | 87.78 | 81.04 | 82.75 | 80.14 | -8.99 |
| | RL w/ Clean&Mem | 69.73 | 77.85 | 63.11 | 82.22 | 79.05 | 77.38 | 74.89 | -14.24 |
| Ref (Carlini et al., 2021) | Before RL | 73.27 | 63.30 | 60.22 | 41.11 | 73.10 | 82.00 | 65.50 | +0.00 |
| | RL w/ Clean | 66.77 | 58.41 | 45.33 | 51.11 | 65.54 | 73.62 | 58.08 | -7.42 |
| | RL w/ Clean&Mem | 62.77 | 54.17 | 43.11 | 50.44 | 65.38 | 72.62 | 58.86 | -6.64 |
| **Reference free** | | | | | | | | | |
| Zlib (Carlini et al., 2021) | Before RL | 49.38 | 58.61 | 73.56 | 43.56 | 50.19 | 45.00 | 53.38 | +0.00 |
| | RL w/ Clean | 45.94 | 54.99 | 66.22 | 35.56 | 46.65 | 39.38 | 48.12 | -5.26 |
| | RL w/ Clean&Mem | 46.04 | 55.30 | 64.89 | 28.89 | 44.87 | 39.00 | 44.74 | -8.64 |
| Min–K%++ (Zhang et al., 2024) | Before RL | 47.57 | 50.90 | 41.90 | 59.11 | 52.27 | 45.88 | 49.61 | +0.00 |
| | RL w/ Clean | 46.25 | 46.78 | 36.67 | 50.89 | 51.35 | 29.62 | 43.59 | -6.02 |
| | RL w/ Clean&Mem | 43.77 | 48.21 | 21.78 | 38.00 | 48.91 | 43.62 | 40.72 | -8.89 |
| Min–K% (Shi et al., 2023) | Before RL | 69.19 | 69.51 | 85.56 | 75.56 | 71.16 | 78.75 | 74.96 | +0.00 |
| | RL w/ Clean | 55.19 | 60.60 | 62.89 | 65.56 | 61.50 | 61.87 | 61.27 | -13.69 |
| | RL w/ Clean&Mem | 53.93 | 59.74 | 59.56 | 62.67 | 57.31 | 59.25 | 58.54 | -16.42 |
| Max–K% (Maini et al., 2024) | Before RL | 64.50 | 64.31 | 65.11 | 81.78 | 67.27 | 76.00 | 69.83 | +0.00 |
| | RL w/ Clean | 53.05 | 51.43 | 49.78 | 50.22 | 51.84 | 57.75 | 52.35 | -17.48 |
| | RL w/ Clean&Mem | 49.03 | 51.04 | 50.00 | 50.00 | 52.34 | 47.50 | 49.99 | -19.84 |
| Loss (Carlini et al., 2021) | Before RL | 69.18 | 69.81 | 86.22 | 77.33 | 70.95 | 79.38 | 75.48 | +0.00 |
| | RL w/ Clean | 55.22 | 60.50 | 62.44 | 65.78 | 61.50 | 62.12 | 61.26 | -14.22 |
| | RL w/ Clean&Mem | 53.99 | 60.01 | 59.33 | 62.67 | 57.40 | 59.38 | 58.80 | -16.68 |

forming RL solely on clean data. Also, the contaminated model, further trained with GRPO, still shows an average performance inflation of 7.14% across six benchmarks and does not fall back as the clean SFT model, shown in Tab. 1. Second, we continue SFT on the SFT contaminated model with an additional 4 epochs on clean data. Fig. 2 and Tab. 23 demonstrate that further SFT is unable to conceal the benchmark contamination, while the pass@1 would continue to rise. Together, these results show that subsequent GRPO training preserves performance inflation while reducing detectable evidence of contamination may have some underlying reasons, rather than simply forgetting the contamination after further training.

## 3.2 THEORETIC ANALYSIS

In this section, we perform theoretical analysis to demonstrate that PPO-style clipping and importance sampling are the root cause of the concealment. Intuitively, the importance sampling and clipping term reweights terms so that the most off-policy trajectories are damped by the clip while typical on-policy ones keep their influence. This reweighting hits non-members more as they have more extreme successes, so clipping cuts misaligned influence and lets ordinary, on-policy successes steer the update. With more headroom, non-member's NLL drops more and the gap contracts.

**Setup.** We denote $\ell(x, y)$ to be the negative log likelihood (NLL) of the current model of generating $y$ given prompt $x$, members as $M$ and non-members as $N$, policy model $\pi_k$ at step $k$. We focus on analyzing the gap $G_k$ of negative log likelihood for members and non-members on correct samples (i.e., $r = 1$), as assessing contamination on erroneous outputs is not especially meaningful. Formally, we can write

$$G_k := \mathbb{E}_{x \sim N} \mathbb{E}_{y \sim \pi_k(\cdot|x)}[\ell_k(x, y) \mid r = 1] - \mathbb{E}_{x \sim M} \mathbb{E}_{y \sim \pi_k(\cdot|x)}[\ell_k(x, y) \mid r = 1] \tag{1}$$

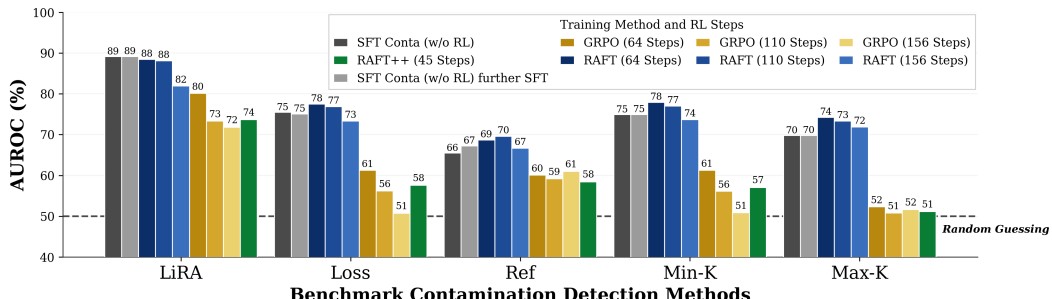

Figure 2: **AUROC (%) trends on SFT contaminated model further trained with different objectives.** While contamination introduced through SFT is initially detectable by existing methods, subsequent RL training with clean samples (e.g., GRPO or RAFT++) consistently degrades detection performance. Moreover, we observe a monotonic decline in detection performance as the number of RL steps increases, and reference-free methods (e.g., Loss, Min-K, and Max-K) already fall into near random guesses (i.e., AUROC≈50%) simply after 156 steps. The base model is Qwen2.5-7B-Instruct. More results of Llama-3.1-8B-Instruct as the base model are shown in Fig.5.

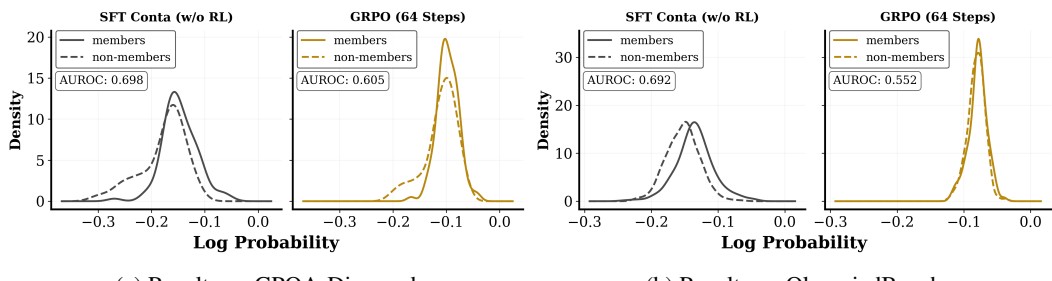

(a) Results on GPQA-Diamond  (b) Results on OlympiadBench

Figure 3: Log-prob distributions for members vs. non-members of **SFT contaminated model before and after RL training.** After further GRPO with clean samples on the SFT contaminated model, the log-prob distributions of members and non-members become increasingly similar. Since many contamination detection methods rely on separability in this space, the shrinking gap explains their degraded effectiveness. More log-prob distributions can be found in Fig. 7, 8, and 9.

If this gap contracts, i.e., $G_{k+1} - G_k < 0$, members and non-members become closer in the NLL sense, making contamination detection harder since many methods (Zhang et al., 2024; Shi et al., 2023; Maini et al., 2024; Carlini et al., 2021) are based on the separation of NLLs. For a fixed prompt $x$, we define the NLL drift as

$$\Delta_x := \mathbb{E}_{\pi_{k+1}}[\ell_{k+1} \mid r = 1, x] - \mathbb{E}_{\pi_k}[\ell_k \mid r = 1, x]. \tag{2}$$

We notice that we can rewrite the NLL gap as

$$G_{k+1} - G_k := \mathbb{E}_{x \in N}[\Delta_x] - \mathbb{E}_{x \in M}[\Delta_x]. \tag{3}$$

In our following analysis, we thus focus on investigating the behavior of $\Delta_x$ on members and non-members. If an algorithm yields on average smaller $\Delta_x$ on non-members, the algorithm should be able to conceal contamination.

**Notations.** At token $t$, let $A_t$ be the method's per token reward/advantage and $w_t$ the weight from importance sampling and clipping. Define

$$A_t^w := w_t A_t, \quad \bar{A}^w(s) := \mathbb{E}_{a \sim \pi_k(\cdot|s)}[A^w(s,a)], \quad \tilde{A}_t^w := A_t^w - \bar{A}^w(s_t) \tag{4}$$

to measure how good a state is compared to the average. In particular, $w_t = \rho_t m_t$ with $\rho_t = \pi_\theta(a_t \mid s_t)/\pi_{\text{old}}(a_t \mid s_t)$ being the importance sampling and $m_t \in \{0, 1\}$ be a mask indicating if the clipping is activated, specifically $m_t = 0$ indicates that there is no gradients from the update. Moreover, we define $p_k(x) = \mathbb{E}_{y \sim \pi_k(\cdot|x)}[r(x,y)]$ to be the overall success rate of the prompt, and a value function as $q_k(s,a) := \Pr(r = 1|s, a)$ and $p_k(s) := \mathbb{E}_{a \sim \pi_k(\cdot|s)}[q_k(s,a)]$ for success rate at that state. And we

Table 3: **AUROC (%)** of detection approach, Loss (Carlini et al., 2021), evaluated on **SFT contaminated model further trained with different RL objectives.** The gray row indicates no ablation on the objective, and ✗ means remove the term from the objective. Δ measures the difference with the SFT contaminated model w/o RL (Tab. 2). RL steps are 64, or the step before the model collapses. The results show that clipping is the main driver for the contraction, which aligns with our theory.

| Training Objectives | Clipping | Olypaid | GPQA | AIME25 | AIME24 | Minerva | AMC23 | Avg. | Δ |
|---|---|---|---|---|---|---|---|---|---|
| RAFT | ✗ | 71.78 | 69.78 | 86.00 | 86.67 | 71.58 | 79.25 | 77.51 | +2.03 |
| RAFT++ | ✓ | 50.43 | 58.45 | 67.56 | 66.67 | 52.84 | 49.50 | 57.58 | -17.91 |
| RAFT++ | ✗ | 69.16 | 73.68 | 74.44 | 76.22 | 71.08 | 81.75 | 74.39 | -1.09 |
| GRPO | ✓ | 55.22 | 60.50 | 62.44 | 65.78 | 61.50 | 62.12 | 61.26 | -14.22 |
| GRPO | ✗ | 68.83 | 70.20 | 80.44 | 73.78 | 68.30 | 78.12 | 73.28 | -2.20 |

define $B(s) = \mathbb{E}_{a \sim \pi}[\rho(s,a)m(s,a)q_k(s,a)]$ and $C(s) = \mathbb{E}_{a \sim \pi}[\rho(s,a)m(s,a)]$. We assume that the RL training is performed on the benchmark data (i.e., training data is the combination of members $M$ and non-members $N$), and it is in a tabular setting for simplicity. Since members have been utilized during training, it is natural to assume the $p_k(s)$ for members are larger than non-members, and the NLL for members is lower than non-members.

**Theorem 3.1.** *For a small natural gradient step with step size $\eta$ on a PPO style loss, we have*

$$\Delta_x = -\eta \underbrace{\mathbb{E}\left[\frac{1}{T}\sum_{t=1}^{T}\tilde{A}_t^w \middle| r=1, x\right]}_{(A)\ \mu(x)} + \eta \underbrace{\text{Cov}\left(\ell_k, \sum_{t=1}^{T}\tilde{A}_t^w\right)}_{(B)\ \text{covariance}\ \beta(x)} + O(\eta^2) \tag{5}$$

The proof can be found in appendix C. Intuitively, $\mu(x)$ measures the average push on the example's NLL from correct trajectories, where $\beta$ serves as a reweighting term accouting for the importance sampling/clipping. Here we consider several instantiations using different algorithms to investigate the core driver for contraction. The training objectives for each algorithm are listed in Appendix B.

**RAFT.** In plain rejection sampling, we have $w_t = 1$ and $A_t = \mathbf{1}\{r = 1\}$, so on correct trajectories

$$\tilde{A}_t^w = 1 - p_k(s_t), \quad \mu^{\text{RAFT}}(x) = \mathbb{E}\left[\frac{1}{T}\sum_{t=1}^{T}\left(1 - p_k(s_t)\right) \middle| r=1, x\right].$$

The covariance term is

$$\beta^{\text{RAFT}}(x) = \text{Cov}\left(\ell_k, \sum(1 - p_k(s_t))\right) = -\text{Cov}\left(\ell_k, \sum p_k(s_t)\right).$$

We note that lower loss $\ell_k$ corresponds to higher probabilities $p_k(s_t)$, and thus $\beta^{\text{RAFT}}(x) > 0$. Moreover, non-members correct trajectories can exhibit much higher variance in loss and probabilities, thus, the $\beta_N$ term is typically larger than $\beta_M$. Consequently,

$$\Delta_N - \Delta_M = -\eta(\mu_N - \mu_M) + \eta(\beta_N - \beta_M),$$

where both gaps $(\mu_N - \mu_M)$ and $(\beta_N - \beta_M)$ are positive. Empirically, the covariance gap offsets the mean gap, yielding $\Delta_N - \Delta_M \geq 0$, i.e., RAFT is unable to conceal contamination evidence.

**RAFT++.** Using the same $A_t = \mathbf{1}\{r = 1\}$, on $r = 1$ paths

$$\tilde{A}_t^w = \rho_t m_t - B_k(s_t), \quad \mu^{\text{RAFT++}}(x) = \mathbb{E}\left[\frac{1}{T}\sum_{t=1}^{T}\left(\rho_t m_t - B_k(s_t)\right) \middle| r=1, x\right].$$

We note that the difference of $\mu$ cannot possibly lead to large deviations between members/non-members as $0 \leq \rho_t m_t \leq 1 + \epsilon$ and $B_k(s_t) \leq 1$ for both groups and the term is normalized by length. For the covariance term though, we have

$$\beta^{\text{RAFT++}}(x) = \text{Cov}\left(\ell_k, \sum(\rho_t m_t - B_k(s_t))\right) = \text{Cov}\left(\ell_k, \sum \rho_t m_t\right) - \text{Cov}\left(\ell_k, \sum B_k(s_t)\right).$$

Compared to RAFT, the new term $\text{Cov}(\ell_k, \sum \rho_t m_t)$ is negative as correct path with higher loss are anomaly and typically got clipped more. Moreover, this is much more prominent in non-members

Table 4: **Pass@1 (%)** of **advanced LRMs before and after SFT contamination with CoT.**

| Models | Olypaid | GPQA | AIME25 | AIME24 | Minerva | AMC23 | Avg. |
|---|---|---|---|---|---|---|---|
| DeepSeek-R1-Distill-Llama-8B | 52.10 | 43.94 | 33.33 | 43.33 | 32.97 | 84.58 | 48.38 |
| ↪ w/ extensive SFT Contamination | 61.83 | 53.16 | 51.67 | 61.67 | 38.74 | 93.75 | 60.14 |
| DeepSeek-R1-Distill-Qwen-7B | 55.70 | 48.65 | 39.26 | 53.70 | 37.25 | 91.94 | 54.42 |
| ↪ w/ extensive SFT Contamination | 58.77 | 50.87 | 42.59 | 58.91 | 40.81 | 90.67 | 57.10 |
| OpenThinker3-7B (15K) | 50.81 | 41.67 | 21.67 | 29.17 | 34.01 | 77.50 | 42.47 |
| ↪ w/ extensive SFT Contamination | 52.74 | 47.64 | 33.33 | 30.48 | 40.56 | 78.70 | 47.25 |
| DeepSeek-R1-Distill-Qwen-14B | 59.89 | 56.69 | 44.44 | 62.78 | 42.28 | 92.92 | 59.83 |
| ↪ w/ extensive SFT Contamination | 66.37 | 62.75 | 64.58 | 77.78 | 46.14 | 97.81 | 69.24 |

due to high variance in correct trajectories loss. The second covariance term, although still negative, are not that significant for non-members compared to members due to an average over all possible actions. Therefore, overall it leads to

$$\Delta_N - \Delta_M = -\eta(\mu_N - \mu_M) + \eta(\beta_N - \beta_M) < 0,$$

i.e., RAFT++ contracts the membership gap. The driver is precisely the PPO-style importance sampling/clipping: it removes the RAFT covariance cancellation by making $\text{Cov}(\ell_k, \sum \rho m)$ non-positive and more negative for non-members.

**GRPO.** Finally, we investigate the GRPO contraction term. To ease the analysis, we consider an idealized setting where we define the advantage term as $A_k(x, y) = r(x, y) - p_k(x)$ with no standard deviation term and $\tilde{A}_t^w = \tilde{A}_t^{\text{RAFT}} - p_k(x)(\rho_t m_t - C(s_t))$. Clearly, we have

$$\mu^{\text{GRPO}}(x) = \mu^{\text{RAFT++}}(x) - p_k(x)\mathbb{E}\left[\frac{1}{T} \sum \left(\rho_t m_t - C_k(s_t)\right) \,\bigg|\, r = 1, x\right]$$

$$\beta^{\text{GRPO}}(x) = \beta^{\text{RAFT++}}(x) - p_k(x)\,\text{Cov}\left(\ell_k, \, \sum(\rho_t m_t - C_k(s_t))\right)$$

By similar argument, we know that the $\mu$ term does not contribute significantly to the concealment. The covariance term can be analyzed similarly to show that the concealment also happen on GRPO thanks to the importance sampling and clipping term.

### 3.2.1 EMPIRICAL SUPPORT

To confirm empirically the prediction of our theoretical results, we evaluate the Loss detector (Carlini et al., 2021) after training with RAFT (Dong et al., 2023)/RAFT++ (Xiong et al., 2025)/GRPO. The overall results can be found in table 3. We conduct the ablation study using Qwen2.5-7B-Instruct as the base model. From the results, there are several observations.

**Effect on detectability.** Under RAFT, the Loss detector (Carlini et al., 2021) performance remains essentially unchanged relative to the SFT contaminated baseline w/o further RL. In contrast, RAFT++ and GRPO (with clipping enabled) produce a sharp drop in detector performance.

**Importance sampling vs. clipping.** The clipping term, often treated purely as a training stabilizer, materially contributes to concealment, as predicted by theory. When we retain importance sampling but *remove clipping* in RAFT++ and GRPO, both algorithms show little to no reduction in Loss-detector performance (Table 3). Intuitively, as the clip threshold $\epsilon \to \infty$, the effective weight satisfies $\sum_t \rho_t m_t \approx T$, and the covariance term in our decomposition tends toward zero for both members and non-members, eliminating the shrinkage effect.

These two observations perfectly reflect our theoretical analysis, empirically validating that the PPO-style importance sampling/clipping term is the key driver behind GRPO contamination concealment. Given that many RL algorithms adopt this term in their objectives, this suggests that a broad class of RL methods may inherently exhibit similar concealment capability.

## 4  CONTAMINATION WITH CoT ON ADVANCED LRMS BARELY LEAVES EVIDENCE (STAGE II: POST-LRM)

**Contamination Setup.** In this setup, we simulate contamination with CoT applied to advanced LRMs at the final stage of training. We use DeepSeek-R1-Distill-Llama-8B, DeepSeek-R1-Distill-Qwen-7B, DeepSeek-R1-Distill-Qwen-14B (Guo et al., 2025), and checkpoints from

Table 5: **AUROC (%)** of contamination detection approaches evaluated on **contaminated, advanced LRMs**. Results demonstrate that even after extensive contamination as the final stage, almost all the detection approaches perform near random guesses (i.e., AUROC≈50%). Each AUROC is averaged over detection scores from 8 rollouts.

| Contamination Detection Methods | Init Models | Olympiad | GPQA | AIME25 | AIME24 | Minerva | AMC23 | Avg. |
|---|---|---|---|---|---|---|---|---|
| **Generation based** | | | | | | | | |
| Verbatim (Wu et al., 2025) | DS Llama-8B | 48.73 | 50.45 | 41.33 | 61.56 | 59.10 | 40.63 | 50.30 |
| | DS Qwen-7B | 46.87 | 55.85 | 60.44 | 68.89 | 56.87 | 50.63 | 56.59 |
| | OpenThink-7B | 43.78 | 55.36 | 60.89 | 56.67 | 51.78 | 42.38 | 51.81 |
| | DS Qwen-14B | 48.51 | 50.73 | 52.38 | 61.11 | 55.18 | 53.79 | 53.62 |
| CDD (Dong et al., 2024) | DS Llama-8B | 51.84 | 53.83 | 60.00 | 53.11 | 58.08 | 57.50 | 55.73 |
| | DS Qwen-7B | 51.46 | 48.29 | 50.00 | 53.78 | 54.71 | 41.00 | 49.87 |
| | OpenThink-7B | 49.98 | 50.23 | 53.31 | 51.24 | 54.52 | 50.44 | 51.62 |
| | DS Qwen-14B | 55.82 | 45.50 | 43.11 | 46.67 | 56.13 | 56.45 | 50.61 |
| **Perturbation based** | | | | | | | | |
| Neighbor (Mattern et al., 2023) | DS Llama-8B | 49.94 | 39.32 | 53.11 | 43.33 | 49.68 | 60.00 | 49.23 |
| | DS Qwen-7B | 52.99 | 40.29 | 62.44 | 49.33 | 55.34 | 54.87 | 52.54 |
| | OpenThink-7B | 53.76 | 42.95 | 34.00 | 42.22 | 52.89 | 51.50 | 46.22 |
| | DS Qwen-14B | 53.20 | 42.23 | 50.89 | 44.00 | 53.46 | 57.38 | 50.19 |
| **Reference based** | | | | | | | | |
| LiRA (Mireshghallah et al., 2022) | DS Llama-8B | 57.92 | 53.01 | 53.56 | 75.33 | 69.44 | 58.75 | 61.34 |
| | DS Qwen-7B | 46.52 | 43.93 | 50.22 | 58.89 | 59.33 | 54.00 | 52.15 |
| | OpenThink-7B | 62.35 | 64.77 | 58.44 | 64.44 | 64.81 | 61.62 | 62.74 |
| | DS Qwen-14B | 59.93 | 55.23 | 75.56 | 66.00 | 66.55 | 70.00 | 65.55 |
| Ref (Carlini et al., 2021) | DS Llama-8B | 53.79 | 46.50 | 46.44 | 64.00 | 63.57 | 51.25 | 54.26 |
| | DS Qwen-7B | 53.30 | 44.37 | 46.89 | 44.22 | 53.09 | 41.75 | 47.27 |
| | OpenThink-7B | 57.34 | 49.86 | 37.56 | 50.44 | 59.30 | 69.12 | 53.94 |
| | DS Qwen-14B | 55.75 | 47.55 | 52.67 | 30.89 | 55.51 | 53.75 | 49.35 |
| **Reference free** | | | | | | | | |
| Zlib (Carlini et al., 2021) | DS Llama-8B | 49.52 | 54.74 | 64.22 | 37.11 | 45.97 | 47.12 | 49.78 |
| | DS Qwen-7B | 46.52 | 57.38 | 64.89 | 36.89 | 43.30 | 42.12 | 48.52 |
| | OpenThink-7B | 45.65 | 55.37 | 74.22 | 36.89 | 43.51 | 36.62 | 48.71 |
| | DS Qwen-14B | 48.12 | 56.71 | 70.44 | 43.56 | 45.92 | 51.50 | 52.71 |
| Min–K%++ (Zhang et al., 2024) | DS Llama-8B | 55.45 | 59.10 | 45.95 | 70.22 | 60.89 | 57.50 | 58.19 |
| | DS Qwen-7B | 48.92 | 56.83 | 48.44 | 59.33 | 51.83 | 62.62 | 54.66 |
| | OpenThink-7B | 51.85 | 58.31 | 66.44 | 55.00 | 49.41 | 41.05 | 53.68 |
| | DS Qwen-14B | 52.44 | 56.72 | 48.44 | 76.44 | 57.39 | 59.62 | 58.51 |
| Min–K% (Shi et al., 2023) | DS Llama-8B | 57.86 | 61.68 | 53.33 | 72.67 | 67.12 | 61.87 | 62.42 |
| | DS Qwen-7B | 49.75 | 53.93 | 51.78 | 61.56 | 54.50 | 56.75 | 54.71 |
| | OpenThink-7B | 53.52 | 57.19 | 60.44 | 57.56 | 54.83 | 47.37 | 55.15 |
| | DS Qwen-14B | 52.77 | 58.08 | 52.44 | 77.33 | 59.43 | 59.62 | 59.95 |
| Max–K% (Maini et al., 2024) | DS Llama-8B | 53.85 | 55.96 | 50.67 | 60.44 | 59.22 | 52.50 | 55.44 |
| | DS Qwen-7B | 49.65 | 50.92 | 40.44 | 73.33 | 54.08 | 56.25 | 54.11 |
| | OpenThink-7B | 55.12 | 58.29 | 46.22 | 79.33 | 54.20 | 59.38 | 58.76 |
| | DS Qwen-14B | 50.43 | 53.89 | 50.00 | 50.00 | 51.08 | 52.50 | 51.32 |
| Loss (Carlini et al., 2021) | DS Llama-8B | 57.91 | 61.78 | 52.89 | 73.56 | 67.00 | 62.38 | 62.59 |
| | DS Qwen-7B | 49.77 | 54.09 | 52.00 | 63.78 | 54.76 | 56.75 | 55.19 |
| | OpenThink-7B | 53.44 | 57.61 | 61.33 | 56.67 | 55.07 | 48.12 | 55.37 |
| | DS Qwen-14B | 52.81 | 58.39 | 52.89 | 77.56 | 59.37 | 60.37 | 60.23 |

OpenThought3 (Guha et al., 2025) as the initial models. We simulate extensive contamination with CoT by applying SFT exclusively on the member data in this section. Additional implementation details are provided in Appendix D.4.

Tab. 4 and 5 show the results of pass@1 on six reasoning benchmarks and AUROC of detection approaches performance (w/ the same detection setup as Stage I), respectively. We observe that:

**Extensive SFT Contamination with CoT results in a huge performance inflation.** As shown in Tab. 4, LRMs can substantially benefit from extensive contamination with CoT. Such inflation enables contaminated LRMs to artificially boost performance in benchmarks and have an overrated rank in the reasoning leaderboard with little extra training cost.

**Extensive contamination with CoT on LRMs barely leaves evidence.** As illustrated in Tab. 5, detection methods, which were effective in contamination introduced when the base model evolves into LRMs, consistently fail under extensive contamination with CoT to LRMs, performing close to random guessing. The previous SOTA approach, LiRA (Mireshghallah et al., 2022), achieves only 58.74% AUROC on average across six benchmarks. Then, we analyze the log prob of member and non-member samples before and after final stage contamination, shown as Fig. 4. After

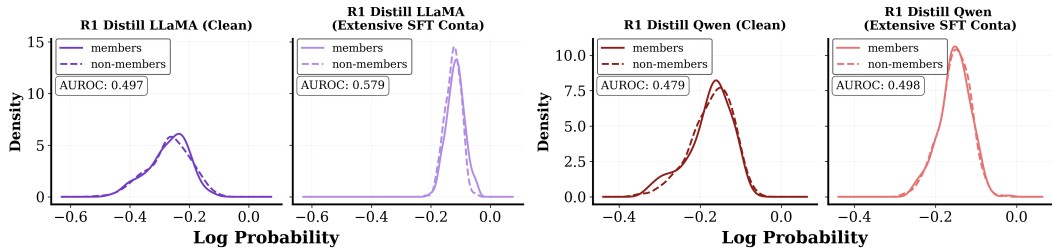

(a) R1 Distill LLaMA results on OlympiadBench     (b) R1 Distill Qwen results on OlympiadBench

Figure 4: Log-prob distributions for members vs. non-members of **advanced LRMs before and after SFT contamination.** After extensive SFT contamination on members, the log prob of both members and non-members increases at a similar margin. More figures are in Fig. 13 and 14.

the extensive SFT contamination with CoT on members, the log prob of both members and non-members increases at a similar margin. This indicates that even without exposure to non-members, contaminated LRMs still have more confidence when responding to unseen samples that share similar distributions to training samples, which also explains why extensive contamination with CoT on LRMs barely leaves evidence. These results suggest that model developers could extensively contaminate their LRMs in the final stage while leaving little detectable evidence.

**Discussion.** Despite Feng et al. (2024) demonstrating that contamination detection could work in non-reasoning model scenarios, the detectors do not have access to the reasoning trajectories used in the LRM contamination scenario, so they have to rely on the generated responses from LRMs. However, LRMs typically possess strong reasoning abilities to output step-by-step long CoT and are difficult to converge on a specific sequence after contamination with long CoT. This may indicate that rather than memorizing specific reasoning trajectories, LRMs internalize the underlying knowledge and reasoning process during the contamination with CoT data, enabling generalization to distributionally similar questions (e.g., non-members). While most detection methods rely on the assumption that contaminated models would achieve lower loss on training sequences (Carlini et al., 2021) or generate less diverse responses for seen questions (Dong et al., 2024) than for unseen ones. Accordingly, these methods rely on a gap in certain metrics (e.g., log-probability, Levenshtein distance, etc.) between trained and unseen samples to determine contamination. Nevertheless, these LRMs could also have lower loss when responding to those unseen samples that share similar distributions to the training set, benefiting from their long CoT ability, as shown in Fig. 4. This confounding factor (i.e, generalization) is not accounted for by existing detection approaches, challenging the assumption that benchmark data contamination is more about memorization (Wu et al., 2025; Morris et al., 2025; Hayes et al., 2025).

## 5 CONCLUSION

We present the first systematic study of benchmark contamination in LRMs, structured around two points where contamination can happen. Our results reveal a critical vulnerability in LRM evaluation: contamination detection methods are fragile and contamination introduced at either stage can be concealed. In Stage I (pre-LRM), while SFT contamination to the base model is initially detectable, contamination evidence can be concealed through subsequent RL training. In Stage II (post-LRM), extensive contamination with CoT on advanced LRMs barely leaves evidence for existing memorization-driven detection methods. Our findings call for an urgent need of protocols that ensure fair evaluations among LRMs. Here, we propose two potential directions to ensure it: (I) Model developers should release more intermediate training checkpoints, enabling the community to better monitor and regulate potential benchmark contamination in each training stage. (II) Researchers working on contamination detections should advance beyond memorization-driven methods and explicitly account for the long CoT reasoning and generalization capacity of LRMs. Despite the assumption that contamination is about memorizing the training data inspires numerous detection methods before the LRM era, it may become outdated right now. Detection approaches that are solely based on log-probs or mitigation approaches such as minor benchmark modifications, are definitely inadequate in this context and risk systematically failing. These findings all highlight the need for new assumptions in contamination detection and the development of contamination-robust evaluation protocols in the LRM setting.

## ACKNOWLEDGEMENTS

The authors thank Rui Yang and Yifan Sun for their helpful feedback. Huan Zhang is supported in part by the AI2050 program at Schmidt Sciences (AI2050 Early Career Fellowship).

## REPRODUCIBLE CLAIM

Our code is available at `https://github.com/ASTRAL-Group/LRM_Conta_Detection_Arena.git`. Detailed implementation of detection approaches, SFT training, and RL training can be found in appendix D.2 and D.4. We also provide the proof of our theory in appendix C.

## ETHICS STATEMENT

We find a new vulnerability of LRM evaluations: contamination introduced at either stage can be concealed. Other than this, we do not have more ethics concerns.

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

APPENDIX CONTENTS

## A    LIMITATIONS

Our work reveals critical vulnerabilities: RL fine-tuning can conceal benchmark contamination when base models evolve into LRMs; contamination with CoT applied to advanced LRMs leaves little evidence detectable by existing memorization-based methods. Although we do not propose a new detection algorithm to mitigate these risks, we reveal that the failure of current detection approaches stems from their reliance on log-probability and on the assumption that training samples consistently incur lower loss than unseen samples. Given the unique characteristics of LRMs, future detection methods must adopt more advanced assumptions to address this fundamental challenge. By highlighting these risks, we aim to spur further research on robust and trustworthy evaluation protocols for LRMs.

## B    ALGORITHMS

**SFT**    Let $\mathcal{D} = \{(q, o)\}$ be a corpus of questions $q$ and responses $o$, where $o$ are distilled from advanced LRMs. Let $\pi_\theta(o \mid q)$ be an autoregressive policy model. The $\pi_\theta$ is then fine-tuned to maximize the log-likelihood over the responses:

$$\mathcal{L}_{\text{SFT}}(\theta) = \mathbb{E}_{(q,o)\sim\mathcal{D}}\left[-\log \pi_\theta(o \mid q)\right]$$

**GRPO**    For each question $q$, GRPO samples a group of outputs $\{o_1, \ldots, o_G\}$ from the old policy $\pi_{\theta_{\text{old}}}$ and then optimizes the policy model $\pi_\theta$ by maximizing the following objective:

$$\mathcal{L}_{\text{GRPO}}(\theta) = \mathbb{E}_{q\sim P(Q), \{o_i\}_{i=1}^G \sim \pi_{\theta_{\text{old}}}(O|q)}$$

$$\left[\frac{1}{G}\sum_{i=1}^{G}\frac{1}{|o_i|}\sum_{t=1}^{|o_i|}\left(\min\left(\frac{\pi_\theta(o_i \mid q)}{\pi_{\theta_{\text{old}}}(o_i \mid q)}A_i, \text{clip}\left(\frac{\pi_\theta(o_i \mid q)}{\pi_{\theta_{\text{old}}}(o_i \mid q)}, 1-\varepsilon, 1+\varepsilon\right)A_i\right) - \beta\,D_{\text{KL}}(\pi_\theta\|\pi_{\text{ref}})\right)\right],$$

$$D_{\text{KL}}(\pi_\theta\|\pi_{\text{ref}}) = \frac{\pi_{\text{ref}}(o_i \mid q)}{\pi_\theta(o_i \mid q)} - \log\frac{\pi_{\text{ref}}(o_i \mid q)}{\pi_\theta(o_i \mid q)} - 1,$$

where $\varepsilon$ and $\beta$ are hyper-parameters. We denote $r \in \{0, 1\}$ as a binary reward function that assigns scalar feedback to a question-output pair. $A_i$ is the advantage, computed using a group of rewards $\{r_1, r_2, \ldots, r_G\}$ corresponding to the outputs within each group:

$$A_i = \frac{r_i - \text{mean}(\{r_1, r_2, \ldots, r_G\})}{\text{std}(\{r_1, r_2, \ldots, r_G\})}.$$

**RAFT**    The RAFT is also referred to as the rejection-sampling fine-tuning. The algorithm consists of two parts:

- **Dataset Collection.** For each question $q$, we sample a group of outputs $\{o_1, \cdots, o_G\}$. For each response $o_i$, we compute the reward $r_i \in \{0, 1\}$. We then retain only the responses that receive a reward of $1$ and put them in a dataset $\mathcal{D}$.

- **Model Fine-tuning.** The current policy $\pi_\theta$ is then fine-tuned to maximize the log-likelihood over the selected dataset:

$$\mathcal{L}_{\text{RAFT}}(\theta) = \mathbb{E}_{(q,o)\sim\mathcal{D}}[-\log \pi_\theta(o \mid q)]$$

**RAFT++**    RAFT++ is a variant of plain RAFT that also introduced importance sampling and clipping. The loss is very similar to GRPO with no KL divergence term except for the advantage calculation. The advantage for RAFT++ will be:

$$A_i = \mathcal{I}(r_i = 1),$$

which indicates essentially that we train only on the positive samples.

## C PROOF

Here we restate our theorem 3.1, and provide a full proof.

**Theorem.** *For a small natural gradient step with step size $\eta$ on a PPO style loss, we have*

$$\Delta_x = -\eta \, \mathbb{E}\left[\frac{1}{T}\sum_{t=1}^{T}\tilde{A}_t^w \,\middle|\, r=1, x\right] + \eta \, \mathrm{Cov}\left(\ell_k, \sum_{t=1}^{T}\tilde{A}_t^w\right) + O(\eta^2) \tag{6}$$

*Proof.* Denote $\mathrm{clip}_\epsilon(\rho) = \min(\max(\rho, 1-\epsilon), 1+\epsilon)$. Consider the optimization problem

$$\mathcal{L}(\theta) = \mathbb{E}\left[\sum_t \min\left(\rho_t(\theta)A_t, \mathrm{clip}_\epsilon(\rho_t(\theta))A_t\right)\right] \tag{7}$$

The gradient of this objective is

$$\nabla_\theta \min\left(\rho_t(\theta)A_t, \mathrm{clip}_\epsilon(\rho_t(\theta))A_t\right) = m_t A_t \rho_t \nabla_\theta \log \pi_\theta(y_t \mid s_t) \tag{8}$$

Let $z_{s,a}$ be the logit parameter for action $a$ at state $s$, so that $\pi_\theta(a \mid s) = \mathrm{softmax}(z_{s,\cdot})_a$. Then

$$\frac{\partial}{\partial z_{s,a}} \log \pi_\theta(y_t \mid s_t) = \mathbf{1}\{s_t = s\}\left(\mathbf{1}\{y_t = a\} - \pi_\theta(a \mid s)\right), \tag{9}$$

and thus

$$\frac{\partial \mathcal{L}}{\partial z_{s,a}} = \mathbb{E}\left[\sum_t \left(\rho_t \, m_t \, A_t\right)\left(\mathbf{1}\{y_t = a\} - \pi_\theta(a \mid s_t)\right)\right]. \tag{10}$$

It is then clear that after one natural-gradient step,

$$z_{s,a}^{k+1} - z_{s,a}^{k} = \eta\left(A^w(s, a) - \bar{A}^w(s)\right) = \eta \tilde{A}^w(s, a), \tag{11}$$

Using a first-order Taylor expansion of $\log \pi$ and equation 11,

$$\log \pi_{k+1}(a \mid s) - \log \pi_k(a \mid s) = \eta \tilde{A}^w(s, a) + O(\eta^2). \tag{12}$$

For a trajectory $y = (y_1, \ldots, y_T)$,

$$\log \frac{\pi_{k+1}(y \mid x)}{\pi_k(y \mid x)} = \sum_{t=1}^{T}\left(\log \pi_{k+1}(y_t \mid s_t) - \log \pi_k(y_t \mid s_t)\right) = \eta \sum_{t=1}^{T}\tilde{A}^w(s_t, y_t) + O(\eta^2). \tag{13}$$

Define

$$\Psi_x(y) := \sum_{t=1}^{T}\tilde{A}^w(s_t, y_t), \quad S_k^w(x, y) := \frac{1}{T}\sum_{t=1}^{T}\tilde{A}^w(s_t, y_t).$$

Then, from $\ell_k(x, y) = -\frac{1}{T}\sum_t \log \pi_k(y_t \mid s_t)$ and equation 12,

$$\ell_{k+1}(x, y) - \ell_k(x, y) = -\frac{1}{T}\sum_{t=1}^{T}\left(\log \pi_{k+1} - \log \pi_k\right) = -\eta \, S_k^w(x, y) + O(\eta^2). \tag{14}$$

Moreover,

$$R_x(y) := \frac{\pi_{k+1}(y \mid x)}{\pi_k(y \mid x)} = \exp\left(\eta \, \Psi_x(y) + O(\eta^2)\right) = 1 + \eta \, \Psi_x(y) + O(\eta^2). \tag{15}$$

Add and subtract $\mathbb{E}_{\pi_{k+1}}[\ell_k \mid r=1, x]$ in $\Delta_x$, we have

$$\Delta_x = \underbrace{\mathbb{E}_{\pi_{k+1}}\left[\ell_{k+1} - \ell_k \mid r=1, x\right]}_{(A)} + \underbrace{\left(\mathbb{E}_{\pi_{k+1}}[\ell_k \mid r=1, x] - \mathbb{E}_{\pi_k}[\ell_k \mid r=1, x]\right)}_{(B)}. \tag{16}$$

Using equation 14 and a change of measure with $R_x$,

$$(A) = \frac{\mathbb{E}_{\pi_k}\big[(\ell_{k+1} - \ell_k)\, R_x \,\big|\, r{=}1, x\big]}{\mathbb{E}_{\pi_k}\big[R_x \,\big|\, r{=}1, x\big]}$$

$$= \frac{\mathbb{E}_{\pi_k}\big[(-\eta S_k^w + O(\eta^2))\,(1 + \eta \Psi_x + O(\eta^2)) \,\big|\, r{=}1, x\big]}{1 + \eta\, \mathbb{E}_{\pi_k}[\Psi_x \mid r{=}1, x] + O(\eta^2)}$$

$$= -\eta \mathbb{E}_{\pi_k}\big[S_k^w \mid r{=}1, x\big] + O(\eta^2)$$

$$= -\eta \mathbb{E}_{\pi_k}\left[\frac{1}{T} \sum_{t=1}^{T} \tilde{A}_t^w \,\middle|\, r{=}1, x\right].$$

where the third equality is simply a first order expansion of division. For term (B), we can compute for any integrable $f$,

$$\mathbb{E}_{\pi_{k+1}}[f \mid r{=}1, x] = \frac{\mathbb{E}_{\pi_k}[f\, R_x \mid r{=}1, x]}{\mathbb{E}_{\pi_k}[R_x \mid r{=}1, x]} = \mathbb{E}_{\pi_k}[f \mid r{=}1, x] + \eta \operatorname{Cov}_{\pi_k(\cdot \mid r{=}1, x)}\big(f, \Psi_x\big) + O(\eta^2),$$

by expanding numerator and denominator using equation 15. Taking $f = \ell_k$ gives

$$(B) = \eta \operatorname{Cov}_{\pi_k(\cdot \mid r{=}1, x)}\big(\ell_k, \Psi_x\big) + O(\eta^2) = \eta \operatorname{Cov}_{\pi_k(\cdot \mid r{=}1, x)}\left(\ell_k, \sum_{t=1}^{T} \tilde{A}_t^w\right). \tag{17}$$

Combining the two part we get our theorem. $\qquad\square$

## D    EXPERIMENT SETUP

### D.1    CONTAMINATION PIPELINES

**SFT Contamination**    We randomly select 10K samples from OpenThoughts3 (Guha et al., 2025) to form the clean SFT training set, following the same ratio for each domain as the original paper, to obtain the best results. For the SFT contaminated training set, we use QwQ-32B (Team, 2025) as an advanced LRM to help distillation for the member set. We adopt the temperature 0.6, top-p value of 0.95, and maximum output token with 32768 for the distillation. We use rejection sampling with 64 rollouts, selecting correct trajectories when available. If none of the 64 rollouts produce a correct answer, we randomly select an incorrect trajectory. We replicate the question in the member set with responses 3 times to create the SFT contamination training set. So the training set consists of 11866 samples. For the Llama-3.1-8B-Instruct, we randomly select 30K samples from OpenThoughts3 (Guha et al., 2025) to form the clean SFT training set, and duplicate memberships 9 times to form the SFT contamination training set.

Table 6: Proportion of questions solved after up to 64 rollouts with QwQ-32B.

| Olypaidbench | GPQA-Diamond | AIME2025 | AIME2024 | Minerva Math | AMC2023 | Avg. |
|---|---|---|---|---|---|---|
| 80.59 | 92.42 | 93.33 | 93.33 | 56.62 | 100.00 | 86.05 |

**RL Contamination**    For RL contamination, we replicate the questions in the member set 2 times, and randomly select 4096 samples from DeepMath-103K (He et al., 2025) as the clean RL training set. We choose GRPO as the RL algorithm and train the model with one epoch.

**Base Model Selection**    We choose Qwen2.5-7B-Instruct (Team, 2024) and Llama-3.1-8B-Instruct Dubey et al. (2024) as the base model, and first train it with SFT and then RL.

### D.2    CONTAMINATION DETECTION METHODS

**Setup**    For input question $q$, response $o$, and model $\pi_\theta$, we define the detection score as $f(q, o, \pi_\theta)$. We treat the contaminated benchmark as the member set and the remaining uncontaminated half as the non-member set. AUROC is computed from the detection scores between the member and non-member set within a benchmark. We define the average log probability of model $\pi_\theta$ generating the response $o$ given the question $q$ as:

$$\phi(q, o, \pi_\theta) \;=\; \frac{1}{|o|} \log \pi_\theta(o \mid q).$$

We provide experiments to illustrate why we choose the log probability on responses for most detection methods in appendix E.2, and assume that if models have seen the questions during the training, they would have more confidence during the generation, thus have higher log probabilities.

#### D.2.1    GENERATION-BASED DETECTION

**Verbatim (Wu et al., 2025)**    Verbatim-based approach (Wu et al., 2025) prompts the model to complete the remaining parts of a question based on partial prefixes. (Wu et al., 2025) uses the partial-prompt completion rate measured by ROUGE-L (Lin, 2004), which calculates the overlap of the longest common subsequence between the generated and reference text. If the model memorizes the question during the training, the partial-prompt completion rate would be higher compared to unseen questions. We use 80% of the original problem to generate a partial completion. Using a lower ratio causes the LRM to answer the question directly instead of continuing the given sequence.

**CDD (Dong et al., 2024)**    CDD measures the variation of the generated responses, and assumes that if the responses share strong similarities, the question is more likely to be contaminated during the training.

$$f(q, o, \pi_\theta) = \sum_{d=0}^{\alpha l} \rho^*(d) = \sum_{d=0}^{\alpha l} \frac{\sum_{i=1}^{|G|} \mathbb{I}(\text{ED}(o_i, o_{\text{temperature}=0}) = d)}{|G|}$$

where ED represents Edit distance (Lcvenshtcin, 1966). We quantify the peakedness of the edit-distance distribution $\rho^*(d)$ by its cumulative mass within a similarity window, i.e., $F(d \leq \alpha l)$, where $\alpha \in [0, 1]$ and $l = \max\{\text{Len}(o) \mid o \in O\}$. In our experiments, we set $\alpha = 0.5$, which performed better than other choices.

### D.2.2 PERTURBATION-BASED DETECTION

**Neighborhood (Mattern et al., 2023)** Neighborhood method calibrates the detection score $\phi(q, o, \pi_\theta)$ with some unseen questions $q'$ that share similar semantics to the original question $q$. We denote GPT-4o as $\mathcal{N}$ to augment the question $q$ and get $q' \in \mathcal{N}(q)$. We use a total of five augmented samples to compute the detection score:

$$f(q, o, \pi_\theta) = \phi(q, o, \pi_\theta) - \phi_{\text{neighbor}}(q', o, \pi_\theta), \qquad \phi_{\text{neighbor}}(q', o, \pi_\theta) = \frac{1}{|\mathcal{N}(q)|} \sum_{q' \in \mathcal{N}(q)} \phi(q', o, \pi_\theta).$$

### D.2.3 REFERENCE-BASED DETECTION

The detector assumes that they could access to the training distribution of the target model $\pi_\theta$, and have a reference model $\pi_{\text{ref}}$ to calibrate the detection scores. We choose bespokelabs/Bespoke-Stratos-7B (Labs, 2024) as the reference model $\pi_{\text{ref}}$ in all the experiments.

**LiRA (Mireshghallah et al., 2022)** LiRA calibrates detection score by dividing.

$$f(q, o, \pi_\theta) = \frac{\phi(q, o, \pi_\theta)}{\phi(q, o, \pi_{\text{ref}})}.$$

**Ref (Carlini et al., 2021)** Ref calibrates detection score by subtraction.

$$f(q, o, \pi_\theta) = \phi(q, o, \pi_\theta) - \phi(q, o, \pi_{\text{ref}}).$$

### D.2.4 REFERENCE-FREE DETECTION

**LOSS (Carlini et al., 2021)** The detection score is as below:

$$f(q, o, \pi_\theta) = \phi(q, o, \pi_\theta).$$

**Zlib (Carlini et al., 2021)** Zlib calibrates the detection score with $\text{Zlib}(o)$, which is a compression-based entropy/length proxy.

$$f(q, o, \pi_\theta) = \frac{\phi(q, o, \pi_\theta)}{\text{Zlib}(o)}.$$

**Min-K% (Shi et al., 2023)** Min-k% compute the detection score on k% tokens with lowest probabilities in the sequence. Following (Shi et al., 2023), we choose k as 20 by default in all the experiments.

$$f(q, o, \pi_\theta) = \frac{1}{|\text{min-}k(o)|} \sum_{i \in \text{min-}k(o)} \left[ \log \pi_\theta(o_i \mid q, o_{<i}) \right].$$

**Min-K++% (Zhang et al., 2024)** Min-k%++ standardizes the log probability before taking Min-K%. $\mu_i$ is the expectation of the next token's log probbaility over the vocabulary $\mathcal{V}$ of the model $\pi_\theta$ given the prefix $q, o_{<i}$, and $\sigma_i$ is the standard deviation. We use k=20 by default.

$$f(q, o, \pi_\theta) = \frac{1}{|\text{min-}k(o)|} \sum_{i \in \text{min-}k(o)} \frac{\log \pi_\theta(o_i \mid q, o_{<i}) - \mu_i}{\sigma_i},$$

$$\mu_i = \mathbb{E}_{v \in \mathcal{V}}[\log \pi_\theta(v \mid q, o_{<i})], \quad \sigma_i = \text{Std}_{v \in \mathcal{V}}[\log \pi_\theta(v \mid q, o_{<i})].$$

**Max-K% (Maini et al., 2024)** Max-k% compute the detection score on k% tokens with largest probabilities in the sequence. We use k=20 by default.

$$f(q, o, \pi_\theta) = \frac{1}{|\text{max-}k(o)|} \sum_{i \in \text{max-}k(o)} \Big[ \log \pi_\theta(o_i \mid q, o_{<i}) \Big].$$

### D.3 DATASETS

**AIME 2024 & 2025** Olympiad-level mathematical-reasoning benchmarks consisting of 30 problems each from the American Invitational Mathematics Examination (AIME) 2024 and 2025.

**AMC 2023** A high school level math benchmark consisting of 40 problems from the 2023 American Mathematics Competitions (AMC).

**GPQA Diamond (Rein et al., 2024)** Graduate level scientific-reasoning, multiple-choice benchmark written by domain experts in biology, physics, and chemistry. The *Diamond* split is the hardest subset, with 198 questions retained only when expert annotators agreed, and non-expert baselines typically fail.

**OlympiadBench (He et al., 2024)** An Olympiad level, bilingual, multimodal benchmark designed to test scientific reasoning in mathematics and physics. We use the English, text-only math subset consisting of 674 competition problems.

**Minerva Math (Lewkowycz et al., 2022)** A challenging quantitative reasoning benchmark derived from Google's Minerva work, consisting of 272 problems.

### D.4 IMPLEMENTATION DETAILS

**SFT Implementation** We use the LLaMA-Factory (Zheng et al., 2024) implementation for our SFT experiments. By default, we adopt the SFT hyperparameters suggested by OpenThought3 (Guha et al., 2025) for medium dataset scales for the scenario that the base model evolves into LRMs, as shown below. To improve training efficiency, we employ FlashAttention-2 (Dao, 2023), DeepSpeed ZeRO-1 (Rasley et al., 2020), Liger kernels (Hsu et al., 2024), and asynchronous activation offloading (Daniel Han & team, 2023).

| Training type | Batch Size | Context Length | LR | Epochs | LR Scheduler | Warmup Ratio | Weight Decay | Training Precision |
|---|---|---|---|---|---|---|---|---|
| Full | 128 | 32,768 | 4e-5 | 5 | cosine | 0.1 | 0 | bf16 |

For the extensive contamination to LRMs in the final stage, we use the hyperparameters as follows:

| Training type | Batch Size | Context Length | LR | Epochs | LR Scheduler | Warmup Ratio | Weight Decay | Training Precision |
|---|---|---|---|---|---|---|---|---|
| Full | 32 | 32,768 | 1e-5 | 7 | cosine | 0.1 | 0 | bf16 |

**RL Implementation** We use Verl (Sheng et al., 2024) implementation for our RL experiments. For RAFT and RAFT++, we follow the implementation of (Xiong et al., 2025). For GRPO, we follow DAPO (Yu et al., 2025) and do not introduce the KL term in our training in all the experiments. The detailed hyperparameters are as follows:

| Training type | Batch Size | Prompt Length | Response Length | $\epsilon$ | LR | Epochs | Rollout Num | Rollout Temp | Training Precision |
|---|---|---|---|---|---|---|---|---|---|
| Full | 64 | 1,024 | 16,384 | 0.2 | 1e-6 | 1 | 4 | 0.6 | bf16 |

**Prompt template** We use a prompt template to enable long CoT during train on both SFT and RL. We use math template to AIME2024 & 2025, AMC2023, OlympiadBench (He et al., 2024), and Minervamath (Lewkowycz et al., 2022). We adapt multiple-choice template to GPQA Diamond (Rein et al., 2024).

**Evaluation and Metric** We evaluate pass@1 and run 10 rollouts on AIME 2024 & 2025, AMC 2023, and 3 rollouts on OlympiadBench, GPQA Diamond, and Minerva Math to compute the pass@1. We use vLLM (Kwon et al., 2023) for the inference. All inference uses the same configurations: temperature=0.6, top_p=0.95, max_new_tokens=32,768.

---

**Reasoning Template for Math**

```
{question}\nPlease reason step by step, and put your final answer
within \boxed{}.
```

---

**Example.** Alice and Bob play the following game. A stack of $n$ tokens
lies before them. The players take turns with Alice going first.
On each turn, the player removes either $1$ token or $4$ tokens from
the stack. Whoever removes the last token wins. Find the number
of positive integers $n$ less than or equal to $2024$ for which there
exists a strategy for Bob that guarantees that Bob will win the game
regardless of Alice's play.\nPlease reason step by step, and put your
final answer within \boxed{}.

---

**Reasoning Template for Multiple Choice Question**

```
Return your final response within \boxed{} and only include
the letter choice (A, B, C, or D) as your final response.
{question}{options}
```

---

**Example.** Return your final response within \boxed{} and only
include the letter choice (A, B, C, or D) as your final response.
trans-cinnamaldehyde was treated with methylmagnesium bromide,
forming product 1.
1 was treated with pyridinium chlorochromate, forming product 2.
3 was treated with (dimethyl(oxo)-l6-sulfaneylidene)methane in DMSO
at elevated temperature, forming product 3.
how many carbon atoms are there in product 3? A) 11, B) 10, C) 12,
D) 14

**Deduplication** We deduplicate our clean training datasets for both RL and SFT against the evaluation benchmarks using 13-gram overlap deduplication to ensure conclusive results.

# E   MORE EXPERIMENT RESULTS

## E.1   MORE RESULTS OF LLAMA-3.1-8B-INSTRUCT (STAGE I: PRE-LRM)

Table 7: **AUROC (%)** of contamination detection approaches evaluated starting from an **SFT-contaminated model w/o RL to subsequently trained with GRPO**. Results demonstrate that after GRPO, AUROC decreases across all the benchmarks and detection approaches. Δ measures the difference with the SFT-contaminated model w/o RL. Higher AUROC, better detection performance. Each AUROC is averaged over detection scores from 8 rollouts. The base model is Llama3.1-8B-Instruct.

| Contamination Detection Methods | Training Stages | Olympiad | GPQA | AIME25 | AIME24 | Minerva | AMC23 | Avg. | Δ |
|---|---|---|---|---|---|---|---|---|---|
| **Generation based** | | | | | | | | | |
| Verbatim (Wu et al., 2025) | Before RL | 48.67 | 65.36 | 51.78 | 60.00 | 56.84 | 52.62 | 55.88 | +0.00 |
| | RL w/ Clean | 49.92 | 50.14 | 57.78 | 58.22 | 57.21 | 57.38 | 55.11 | -0.77 |
| | RL w/ Clean&Mem | 48.42 | 48.21 | 54.89 | 56.22 | 57.83 | 58.00 | 53.93 | -1.95 |
| CDD (Dong et al., 2024) | Before RL | 56.72 | 54.55 | 53.33 | 55.11 | 60.03 | 66.75 | 57.75 | +0.00 |
| | RL w/ Clean | 53.14 | 55.69 | 60.00 | 46.22 | 56.31 | 64.00 | 55.89 | -1.86 |
| | RL w/ Clean&Mem | 57.24 | 49.31 | 53.11 | 43.78 | 57.52 | 63.50 | 54.08 | -3.67 |
| **Perturbation based** | | | | | | | | | |
| Neighbor (Mattern et al., 2023) | Before RL | 51.64 | 40.46 | 62.44 | 41.11 | 53.86 | 63.62 | 52.19 | +0.00 |
| | RL w/ Clean | 52.48 | 37.07 | 55.78 | 39.33 | 50.32 | 65.00 | 50.00 | -2.19 |
| | RL w/ Clean&Mem | 52.34 | 39.76 | 49.78 | 37.11 | 54.44 | 70.25 | 50.61 | -1.58 |
| **Reference based** | | | | | | | | | |
| LiRA (Mireshghallah et al., 2022) | Before RL | 80.50 | 70.86 | 98.22 | 93.78 | 81.54 | 98.87 | 87.30 | +0.00 |
| | RL w/ Clean | 68.95 | 65.57 | 51.33 | 41.11 | 73.99 | 80.75 | 63.62 | -23.68 |
| | RL w/ Clean&Mem | 68.72 | 65.74 | 90.44 | 89.11 | 72.32 | 94.12 | 80.08 | -7.22 |
| Ref (Carlini et al., 2021) | Before RL | 60.21 | 48.62 | 66.00 | 40.89 | 60.47 | 83.50 | 59.95 | +0.00 |
| | RL w/ Clean | 55.27 | 47.41 | 40.67 | 32.44 | 52.31 | 53.25 | 46.89 | -13.06 |
| | RL w/ Clean&Mem | 55.49 | 50.37 | 46.00 | 31.11 | 52.98 | 62.50 | 49.74 | -10.21 |
| **Reference free** | | | | | | | | | |
| Zlib (Carlini et al., 2021) | Before RL | 49.46 | 56.21 | 88.44 | 72.00 | 47.04 | 60.25 | 62.23 | +0.00 |
| | RL w/ Clean | 46.53 | 56.37 | 66.67 | 50.00 | 45.91 | 41.38 | 51.14 | -11.09 |
| | RL w/ Clean&Mem | 47.29 | 52.29 | 66.22 | 60.89 | 45.63 | 46.00 | 53.05 | -9.18 |
| Min–K%++ (Zhang et al., 2024) | Before RL | 48.88 | 49.33 | 41.43 | 39.11 | 51.67 | 30.63 | 43.51 | +0.00 |
| | RL w/ Clean | 46.67 | 47.61 | 29.33 | 24.22 | 59.85 | 34.38 | 40.34 | -3.17 |
| | RL w/ Clean&Mem | 48.08 | 50.65 | 36.22 | 28.44 | 59.27 | 46.63 | 44.88 | +1.37 |
| Min–K% (Shi et al., 2023) | Before RL | 68.90 | 72.17 | 98.22 | 99.56 | 76.16 | 94.50 | 84.92 | +0.00 |
| | RL w/ Clean | 59.11 | 69.85 | 60.44 | 67.56 | 73.89 | 75.87 | 67.79 | -17.13 |
| | RL w/ Clean&Mem | 60.41 | 63.87 | 87.33 | 89.56 | 73.57 | 84.50 | 76.54 | -8.38 |
| Max–K% (Maini et al., 2024) | Before RL | 63.60 | 61.58 | 92.44 | 89.33 | 69.12 | 91.50 | 77.93 | +0.00 |
| | RL w/ Clean | 50.59 | 52.50 | 50.00 | 50.00 | 50.37 | 52.50 | 50.99 | -26.94 |
| | RL w/ Clean&Mem | 50.91 | 52.24 | 49.11 | 49.11 | 54.86 | 61.75 | 53.00 | -24.93 |
| Loss (Carlini et al., 2021) | Before RL | 69.44 | 72.88 | 98.67 | 99.56 | 76.91 | 94.50 | 85.33 | +0.00 |
| | RL w/ Clean | 59.18 | 69.84 | 60.00 | 68.00 | 74.02 | 76.75 | 67.97 | -17.36 |
| | RL w/ Clean&Mem | 60.37 | 63.92 | 89.33 | 90.44 | 74.18 | 84.62 | 77.14 | -8.19 |

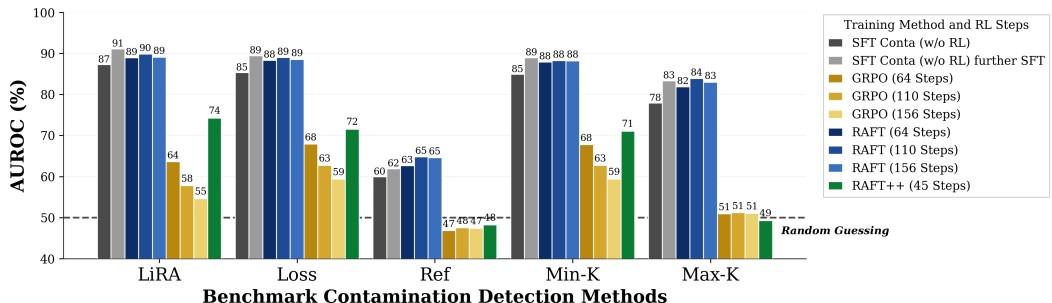

Figure 5: **AUROC (%) trends on SFT contaminated model further trained with different objectives.** While contamination introduced through SFT is initially detectable by existing methods, subsequent RL training with clean samples (e.g., GRPO or RAFT++) consistently degrades detection performance. Moreover, we observe a monotonic decline in detection performance as the number of RL steps increases, and reference-free methods (e.g., Loss, Min-K, and Max-K) already fall into near random guesses (i.e., AUROC≈50%) simply after 156 steps. The base model is Llama3.1-8B-Instruct.

### E.2 AUROC ON QUESTION, QUESTION+RESPONSE, THINKING PROCESS, AND NON-THINKING PROCESS (STAGE I: PRE-LRM)

We compare the AUROC computed on response tokens with computed on the question tokens, whole tokens, thinking tokens, and non-thinking tokens in the response. As shown in Tab. 8 and 11, none of the approach outperform the AUROC computed on the response tokens. Thus, we choose to compute all the detection scores on the response if applicable. The base model here is Qwen2.5-7B-Instruct.

Table 8: AUROC (%) of contamination detection approaches using **question tokens** to compute the detection score, evaluated on the SFT-contaminated model w/o RL. Δ measures the difference with the **reponse tokens** as signal (Tab. 2). Results demonstrate that question tokens are not suitable for detection in the LRM contamination setting, compared with using response tokens.

| Contamination Detection Methods | Olympiad | GPQA | AIME25 | AIME24 | Minerva | AMC23 | Avg. | Δ |
|---|---|---|---|---|---|---|---|---|
| **Generation base** | | | | | | | | |
| Verbatim (Wu et al., 2025) | 47.58 | 49.86 | 47.56 | 53.56 | 52.52 | 65.50 | 52.76 | +0.00 |
| **Perturbation base** | | | | | | | | |
| Neighbor (Mattern et al., 2023) | 47.04 | 56.10 | 44.22 | 32.44 | 49.99 | 54.62 | 47.40 | -3.31 |
| **Reference base** | | | | | | | | |
| LiRA (Mireshghallah et al., 2022) | 49.78 | 43.12 | 42.22 | 46.00 | 48.77 | 55.00 | 47.48 | -41.65 |
| Ref (Carlini et al., 2021) | 49.80 | 48.47 | 40.22 | 47.56 | 49.38 | 53.88 | 48.22 | -17.28 |
| **Reference free** | | | | | | | | |
| Zlib (Carlini et al., 2021) | 48.15 | 50.78 | 60.00 | 40.89 | 54.23 | 52.75 | 51.13 | -2.25 |
| Min–K%++ (Zhang et al., 2024) | 44.33 | 50.53 | 32.00 | 36.89 | 49.67 | 34.25 | 41.28 | -8.33 |
| Min–K% (Shi et al., 2023) | 46.68 | 51.91 | 53.33 | 43.56 | 54.13 | 44.38 | 49.00 | -15.96 |
| Max–K% (Maini et al., 2024) | 49.39 | 54.58 | 71.56 | 55.33 | 54.98 | 66.25 | 58.68 | -11.15 |
| Loss (Carlini et al., 2021) | 47.38 | 54.03 | 58.44 | 46.00 | 55.61 | 52.00 | 52.24 | -23.24 |

Table 9: AUROC (%) of contamination detection approaches using **question + response tokens** to compute the detection score, evaluated on the SFT-contaminated model w/o RL. $\Delta$ measures the difference with **response tokens** as signal (Tab. 2). Results demonstrate that considering both question and response tokens when computing detection scores actually harms the AUROC.

| Contamination Detection Methods | Olympiad | GPQA | AIME25 | AIME24 | Minerva | AMC23 | Avg. | $\Delta$ |
|---|---|---|---|---|---|---|---|---|
| **Perturbation base** | | | | | | | | |
| Neighbor (Mattern et al., 2023) | 54.56 | 43.42 | 48.89 | 38.44 | 56.09 | 61.50 | 50.48 | -0.23 |
| **Reference base** | | | | | | | | |
| LiRA (Mireshghallah et al., 2022) | 79.31 | 72.49 | 89.56 | 71.11 | 66.07 | 89.00 | 77.92 | -11.21 |
| Ref (Carlini et al., 2021) | 72.95 | 64.91 | 60.89 | 40.00 | 72.36 | 82.25 | 65.56 | +0.06 |
| **Reference free** | | | | | | | | |
| Zlib (Carlini et al., 2021) | 46.96 | 54.24 | 69.33 | 41.11 | 43.31 | 42.00 | 49.49 | -3.89 |
| Min–K%++ (Zhang et al., 2024) | 40.49 | 51.08 | 36.19 | 48.67 | 39.61 | 30.50 | 41.09 | -8.52 |
| Min–K% (Shi et al., 2023) | 56.46 | 56.73 | 85.78 | 73.78 | 48.48 | 58.13 | 63.23 | -1.73 |
| Max–K% (Maini et al., 2024) | 64.43 | 63.78 | 64.22 | 81.78 | 66.09 | 76.50 | 69.47 | -0.36 |
| Loss (Carlini et al., 2021) | 59.18 | 58.44 | 87.11 | 76.44 | 49.75 | 63.12 | 65.67 | -9.81 |

Table 10: AUROC (%) of contamination-detection methods using **reasoning tokens only** (tokens inside `<think></think>`) to compute the detection score, evaluated on the SFT-contaminated model w/o RL. The last column, $\Delta$, reports the difference relative to using **the entire response** as the signal (Tab. 2). Results show only minor differences between reasoning token only and whole-response signals, so we use the entire response in the main analysis.

| Contamination Detection Methods | Olympiad | GPQA | AIME25 | AIME24 | Minerva | AMC23 | Avg. | $\Delta$ |
|---|---|---|---|---|---|---|---|---|
| **Reference free** | | | | | | | | |
| Zlib (Carlini et al., 2021) | 49.67 | 59.60 | 70.89 | 35.24 | 48.60 | 46.00 | 51.67 | -0.93 |
| Min–K% (Shi et al., 2023) | 70.76 | 66.46 | 82.89 | 74.52 | 70.34 | 82.25 | 74.54 | +0.69 |
| Max–K% (Maini et al., 2024) | 67.40 | 62.00 | 66.67 | 82.38 | 65.87 | 79.00 | 70.55 | +0.82 |
| Loss (Carlini et al., 2021) | 70.81 | 66.75 | 84.89 | 74.52 | 70.06 | 82.75 | 74.96 | +1.21 |

Table 11: AUROC (%) of contamination-detection methods using **non-reasoning tokens only** (tokens after `</think>`) to compute the detection score, evaluated on the SFT-contaminated model w/o RL. The last column, $\Delta$, reports the difference relative to using **the entire response** as the signal (Tab. 2). Results show that using non-reasoning tokens degrades performance compared with whole-response signals, so we use the entire response in the main analysis.

| Contamination Detection Methods | Olympiad | GPQA | AIME25 | AIME24 | Minerva | AMC23 | Avg. | $\Delta$ |
|---|---|---|---|---|---|---|---|---|
| **Reference free** | | | | | | | | |
| Zlib (Carlini et al., 2021) | 59.73 | 51.49 | 49.11 | 61.67 | 63.37 | 74.25 | 59.94 | +7.34 |
| Min–K% (Shi et al., 2023) | 59.26 | 54.82 | 48.22 | 80.00 | 64.8 | 76.5 | 63.93 | -9.92 |
| Max–K% (Maini et al., 2024) | 60.86 | 54.93 | 55.33 | 69.52 | 59.76 | 77.5 | 62.98 | -6.75 |
| Loss (Carlini et al., 2021) | 59.28 | 54.78 | 48.44 | 80.00 | 64.74 | 76.5 | 63.96 | -9.79 |

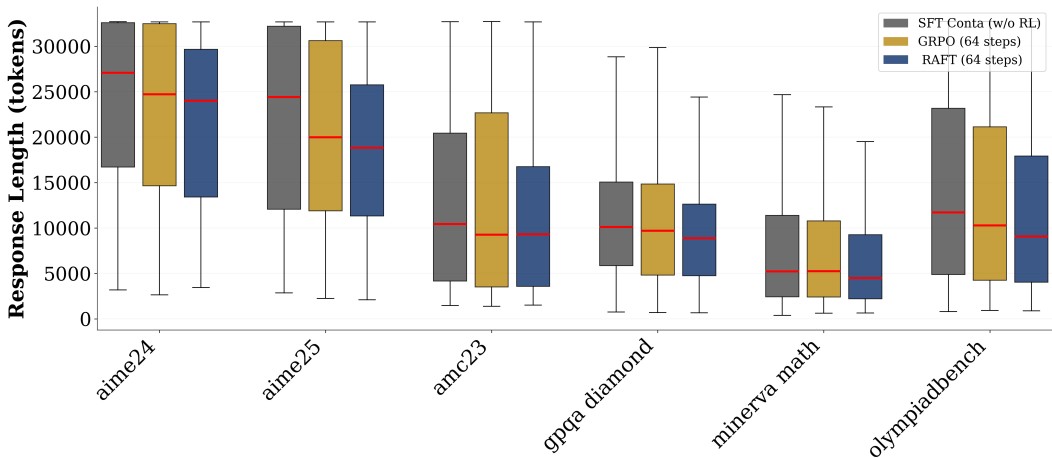

Figure 6: Average response length for each question across different benchmarks.

### E.3 RESPONSE LENGTH ANALYSIS (STAGE I: PRE-LRM)

We visualize the average response length per question across different benchmarks in Fig. 6. Here, the base model is Qwen2.5-7B-Instruct. Although the average response length does not change too much after GRPO training, several detection approaches already exhibit a substantial performance drop, as shown in Tab. 2. In contrast, for RAFT, even though the response lengths remain similar to those after GRPO, no concealment effect is observed. These results suggest that response length is not the key factor; instead, the entropy of the model's output appears to drive the concealment phenomenon.

### E.4 Log-probability distribution before and after RL (Stage I: pre-LRM)

We provide the log probability distribution for members vs. non-members in diverse scenarios. Fig. 7, 8, and 9 demonstrate that further SFT and RAFT are unable to contract the distribution for members and non-members, while GRPO and RAFT++ could conceal the contamination evidence due to the PPO-style importance sampling/clipping term in the training objective. The base model here is Qwen2.5-7B-Instruct.

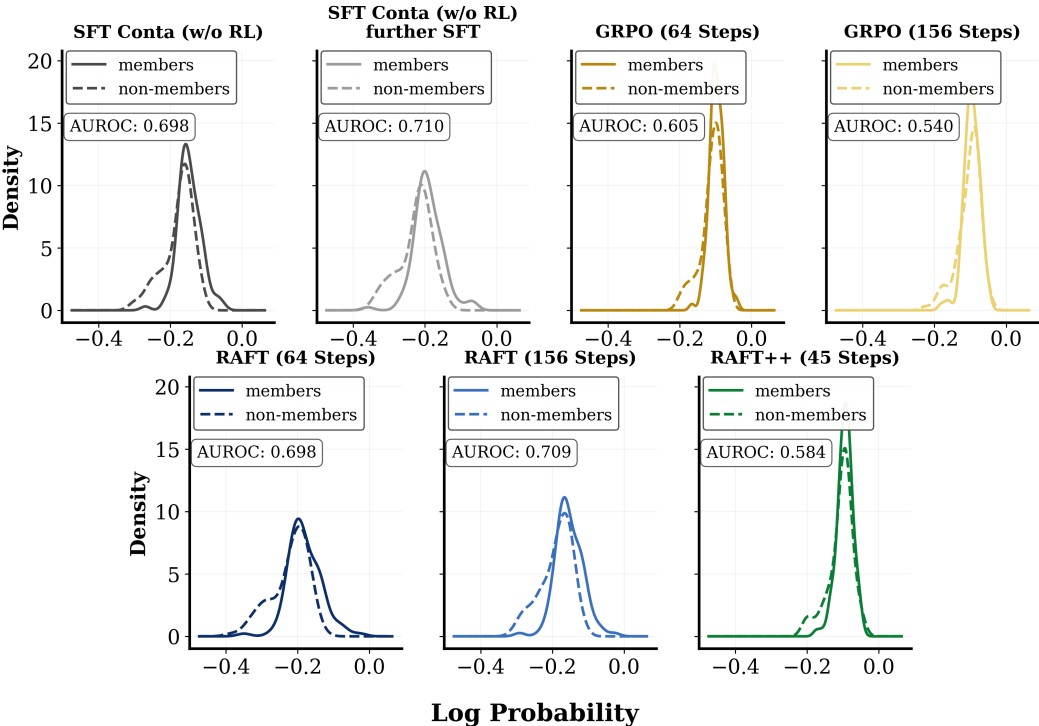

Figure 7: Log-prob distributions for members vs. non-members of **SFT contaminated model before and after RL training** on GPQA-Diamond. With additional GRPO or RAFT++ training on clean samples, the member and non-member log-probability distributions become increasingly similar. Since many contamination detection methods rely on separability in this space, the shrinking gap explains their degraded effectiveness. In contrast, further RAFT training does not induce the earlier distribution collapse; as we explain in Sec. 3.2, the absence of a clipping term prevents it. Likewise, additional SFT does not collapse the membership distributions.

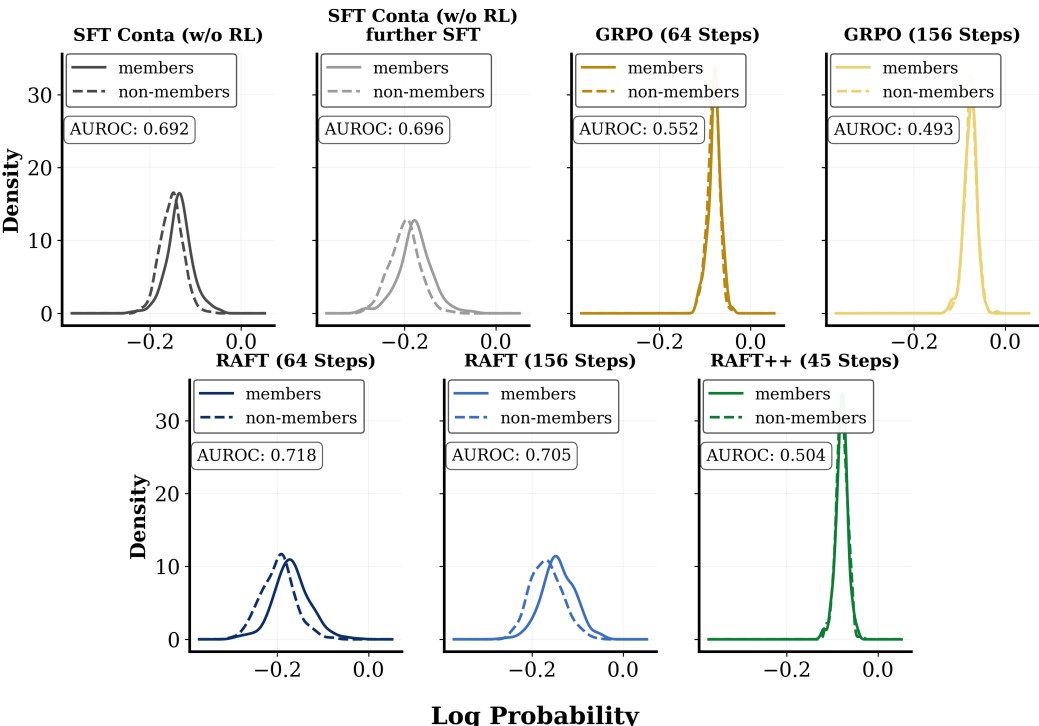

Figure 8: Log-prob distributions for members vs. non-members of **SFT contaminated model before and after RL training** on OlympiadBench. With additional GRPO or RAFT++ training on clean samples, the member and non-member log-probability distributions become increasingly similar. Since many contamination detection methods rely on separability in this space, the shrinking gap explains their degraded effectiveness. In contrast, further RAFT training does not induce the earlier distribution collapse; as we explain in Sec. 3.2, the absence of a clipping term prevents it. Likewise, additional SFT does not collapse the membership distributions.

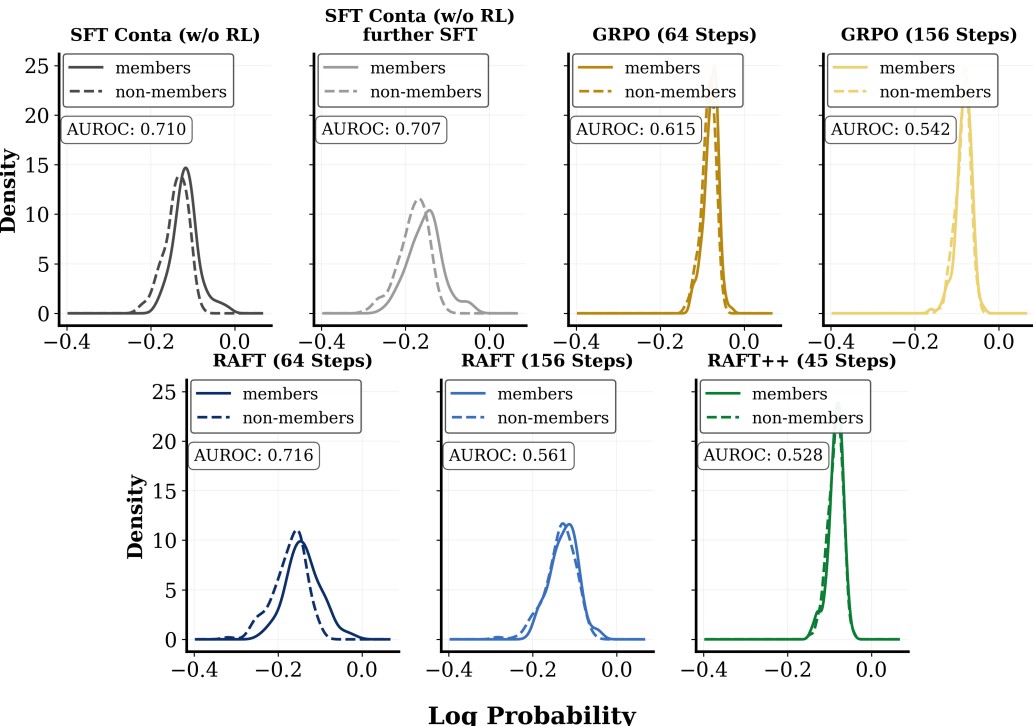

Figure 9: Log-prob distributions for members vs. non-members of **SFT contaminated model before and after RL training** on Minerva Math. With additional GRPO or RAFT++ training on clean samples, the member and non-member log-probability distributions become increasingly similar. Since many contamination detection methods rely on separability in this space, the shrinking gap explains their degraded effectiveness. In contrast, further RAFT training does not induce the earlier distribution collapse; as we explain in Sec. 3.2, the absence of a clipping term prevents it. Likewise, additional SFT does not collapse the membership distributions.

### E.5 AUROC FOR DIFFERENT TRAINING STEPS (STAGE I: PRE-LRM)

We provide complete results of AUROC in different RL training steps. As shown in Tab. 12 and 13, we observe a monotonic decline in detection performance as the number of RL steps increases when using GRPO. While even with 156 steps in RAFT, there is no sign of AUROC decline. These results perfectly validate our theoretical analysis that RAFT is unable to conceal contamination, while GRPO could.

Table 12: AUROC (%) of contamination detection approaches evaluated starting from an SFT-contaminated model w/o RL to subsequently trained with **GRPO in different steps**. $\Delta$ measures the difference with the SFT contaminated model w/o RL (Tab. 2). The results demonstrate that additional GRPO steps better conceal the contamination evidence. The base model here is Qwen2.5-7B-Instruct.

| Contamination Detection Methods | Steps | Olympiad | GPQA | AIME25 | AIME24 | Minerva | AMC23 | Avg. | $\Delta$ |
|---|---|---|---|---|---|---|---|---|---|
| **Generation base** | | | | | | | | | |
| Verbatim (Wu et al., 2025) | 64 | 45.46 | 50.49 | 48.89 | 57.56 | 52.74 | 59.63 | 52.46 | -0.30 |
| | 110 | 45.27 | 52.32 | 52.44 | 50.22 | 50.86 | 60.88 | 52.00 | -0.76 |
| | 156 | 46.51 | 51.09 | 53.33 | 53.56 | 52.84 | 58.50 | 52.64 | -0.12 |
| CDD (Dong et al., 2024) | 64 | 55.47 | 51.08 | 43.33 | 60.00 | 60.18 | 62.00 | 55.34 | -0.45 |
| | 110 | 54.90 | 54.20 | 49.33 | 59.11 | 53.48 | 58.88 | 54.98 | -0.81 |
| | 156 | 53.99 | 43.42 | 34.22 | 70.89 | 56.32 | 62.38 | 53.54 | -2.25 |
| **Perturbation base** | | | | | | | | | |
| Neighbor (Mattern et al., 2023) | 64 | 54.10 | 39.68 | 50.67 | 44.22 | 53.42 | 60.5 | 50.43 | -0.28 |
| | 110 | 53.93 | 40.93 | 43.11 | 46.22 | 54.27 | 59.62 | 49.68 | -1.03 |
| | 156 | 53.32 | 41.07 | 47.11 | 38.67 | 54.49 | 60.75 | 49.24 | -1.47 |
| **Reference base** | | | | | | | | | |
| LiRA (Mireshghallah et al., 2022) | 64 | 74.41 | 84.65 | 70.22 | 87.78 | 81.04 | 82.75 | 80.14 | -8.99 |
| | 110 | 67.87 | 79.80 | 60.22 | 78.89 | 81.49 | 71.88 | 73.36 | -15.77 |
| | 156 | 66.41 | 77.28 | 54.22 | 80.22 | 77.62 | 75.13 | 71.81 | -17.32 |
| Ref (Carlini et al., 2021) | 64 | 66.77 | 58.41 | 45.33 | 51.11 | 65.54 | 73.62 | 60.13 | -5.37 |
| | 110 | 64.92 | 54.70 | 44.44 | 52.22 | 66.47 | 72.37 | 59.19 | -6.31 |
| | 156 | 62.20 | 57.30 | 50.89 | 55.78 | 64.62 | 75.25 | 61.01 | -4.49 |
| **Reference free** | | | | | | | | | |
| Zlib (Carlini et al., 2021) | 64 | 45.94 | 54.99 | 66.22 | 35.56 | 46.65 | 39.38 | 48.12 | -5.26 |
| | 110 | 44.98 | 56.54 | 64.22 | 39.11 | 44.93 | 38.38 | 48.03 | -5.35 |
| | 156 | 44.82 | 53.21 | 63.11 | 41.33 | 46.43 | 39.12 | 48.00 | -5.38 |
| Min–K%++ (Zhang et al., 2024) | 64 | 46.25 | 46.78 | 36.67 | 50.89 | 51.35 | 29.62 | 43.59 | -6.02 |
| | 110 | 43.17 | 44.69 | 28.33 | 50.00 | 49.44 | 46.25 | 43.65 | -5.96 |
| | 156 | 46.12 | 44.86 | 32.44 | 44.00 | 53.39 | 35.38 | 42.70 | -6.91 |
| Min–K% (Shi et al., 2023) | 64 | 55.19 | 60.60 | 62.89 | 65.56 | 61.50 | 61.87 | 61.27 | -13.69 |
| | 110 | 49.98 | 58.49 | 61.78 | 61.33 | 56.20 | 49.00 | 56.13 | -18.83 |
| | 156 | 49.17 | 54.14 | 44.67 | 54.44 | 53.97 | 48.75 | 50.86 | -24.10 |
| Max–K% (Maini et al., 2024) | 64 | 53.05 | 51.43 | 49.78 | 50.22 | 51.84 | 57.75 | 52.35 | -17.48 |
| | 110 | 51.02 | 51.39 | 49.78 | 53.33 | 51.88 | 47.50 | 50.82 | -19.01 |
| | 156 | 49.81 | 53.31 | 50.00 | 50.22 | 51.52 | 55.00 | 51.64 | -18.19 |
| Loss (Carlini et al., 2021) | 64 | 55.22 | 60.50 | 62.44 | 65.78 | 61.50 | 62.12 | 61.26 | -14.22 |
| | 110 | 50.17 | 58.25 | 60.67 | 62.89 | 56.47 | 49.12 | 56.26 | -19.22 |
| | 156 | 49.32 | 54.04 | 44.22 | 54.22 | 54.20 | 48.50 | 50.75 | -24.73 |

Table 13: AUROC (%) of contamination detection approaches evaluated starting from an SFT-contaminated model w/o RL to subsequently trained with **RAFT in different steps**. $\Delta$ measures the difference with the SFT contaminated model w/o RL (Tab. 2). Results demonstrate that even with more RL steps, RAFT is unable to conceal the contamination evidence. The base model here is Qwen2.5-7B-Instruct.

| Contamination Detection Methods | Steps | Olympiad | GPQA | AIME25 | AIME24 | Minerva | AMC23 | Avg. | $\Delta$ |
|---|---|---|---|---|---|---|---|---|---|
| **Generation base** | | | | | | | | | |
| Verbatim (Wu et al., 2025) | 64 | 45.43 | 51.19 | 59.33 | 60.67 | 51.48 | 61.13 | 54.87 | +2.11 |
| | 110 | 45.67 | 51.24 | 58.67 | 58.67 | 52.33 | 57.50 | 54.01 | +1.25 |
| | 156 | 45.15 | 52.16 | 59.11 | 59.33 | 51.28 | 61.25 | 54.71 | +1.95 |
| CDD (Dong et al., 2024) | 64 | 57.85 | 53.55 | 46.89 | 60.22 | 59.63 | 66.62 | 57.46 | +1.67 |
| | 110 | 55.97 | 52.86 | 43.11 | 52.89 | 56.45 | 58.25 | 53.26 | -2.53 |
| | 156 | 55.59 | 52.94 | 32.22 | 53.11 | 55.84 | 67.62 | 52.89 | -2.90 |
| **Perturbation base** | | | | | | | | | |
| Neighbor (Mattern et al., 2023) | 64 | 55.44 | 39.63 | 50.22 | 43.11 | 53.15 | 58.88 | 50.07 | -0.64 |
| | 110 | 55.06 | 38.47 | 46.89 | 40.67 | 54.20 | 63.38 | 49.78 | -0.93 |
| | 156 | 55.06 | 39.19 | 40.89 | 41.78 | 49.66 | 57.12 | 47.28 | -3.43 |
| **Reference base** | | | | | | | | | |
| LiRA (Mireshghallah et al., 2022) | 64 | 85.85 | 85.69 | 98.22 | 84.22 | 84.86 | 91.75 | 88.43 | -0.70 |
| | 110 | 84.32 | 85.76 | 94.44 | 88.44 | 84.55 | 91.25 | 88.13 | -1.00 |
| | 156 | 83.32 | 86.57 | 94.44 | 82.00 | 58.73 | 86.63 | 81.95 | -7.18 |
| Ref (Carlini et al., 2021) | 64 | 76.02 | 65.94 | 74.67 | 46.00 | 73.99 | 75.62 | 68.71 | +3.21 |
| | 110 | 76.41 | 63.76 | 71.11 | 50.22 | 74.50 | 81.75 | 69.63 | +4.13 |
| | 156 | 75.92 | 64.23 | 67.33 | 54.67 | 56.73 | 81.38 | 66.71 | +1.21 |
| **Reference free** | | | | | | | | | |
| Zlib (Carlini et al., 2021) | 64 | 51.88 | 59.73 | 76.00 | 52.22 | 52.40 | 47.75 | 56.66 | +3.28 |
| | 110 | 52.39 | 62.20 | 76.22 | 49.78 | 74.51 | 45.88 | 56.29 | +2.91 |
| | 156 | 53.04 | 62.35 | 79.78 | 47.78 | 52.58 | 50.37 | 57.65 | +4.27 |
| Min–K%++ (Zhang et al., 2024) | 64 | 51.14 | 54.13 | 51.78 | 62.00 | 59.86 | 52.75 | 55.28 | +5.67 |
| | 110 | 52.30 | 55.56 | 56.67 | 62.14 | 55.00 | 49.50 | 55.20 | +5.59 |
| | 156 | 51.26 | 59.72 | 61.19 | 58.44 | 52.53 | 63.12 | 57.71 | +8.10 |
| Min–K% (Shi et al., 2023) | 64 | 72.28 | 69.56 | 88.44 | 86.67 | 71.63 | 78.88 | 77.91 | +2.95 |
| | 110 | 71.44 | 73.69 | 78.67 | 90.22 | 70.45 | 77.62 | 77.02 | +2.06 |
| | 156 | 70.76 | 71.12 | 84.00 | 80.67 | 56.12 | 79.38 | 73.68 | -1.28 |
| Max–K% (Maini et al., 2024) | 64 | 68.11 | 68.37 | 68.44 | 90.67 | 69.21 | 80.50 | 74.22 | +4.39 |
| | 110 | 67.97 | 68.16 | 65.78 | 92.44 | 69.00 | 76.75 | 73.35 | +3.52 |
| | 156 | 69.14 | 68.30 | 69.33 | 90.67 | 58.06 | 76.00 | 71.92 | +2.09 |
| Loss (Carlini et al., 2021) | 64 | 71.78 | 69.78 | 86.00 | 86.67 | 71.58 | 79.25 | 77.51 | +2.03 |
| | 110 | 71.09 | 73.54 | 78.44 | 91.11 | 70.13 | 77.12 | 76.90 | +1.42 |
| | 156 | 70.49 | 70.95 | 83.11 | 80.44 | 56.07 | 79.12 | 73.36 | -2.12 |

Table 14: AUROC (%) of contamination detection approaches evaluated starting from an SFT-contaminated model w/o RL to subsequently trained with **GRPO in different steps**. $\Delta$ measures the difference with the SFT contaminated model w/o RL (Tab. 7). The results demonstrate that additional GRPO steps better conceal the contamination evidence. The base model here is Llama-3.1-8B-Instruct.

| Contamination Detection Methods | Steps | Olympiad | GPQA | AIME25 | AIME24 | Minerva | AMC23 | Avg. | $\Delta$ |
|---|---|---|---|---|---|---|---|---|---|
| **Generation base** | | | | | | | | | |
| Verbatim (Wu et al., 2025) | 64 | 49.92 | 50.14 | 57.78 | 58.22 | 57.21 | 57.38 | 55.11 | -0.77 |
| | 110 | 49.53 | 51.25 | 54.22 | 60.67 | 54.73 | 53.12 | 53.92 | -1.96 |
| | 156 | 49.45 | 52.11 | 51.11 | 60.44 | 57.02 | 47.12 | 52.88 | -3.00 |
| CDD (Dong et al., 2024) | 64 | 53.14 | 55.69 | 60.00 | 46.22 | 56.31 | 64.00 | 55.89 | -1.89 |
| | 110 | 53.48 | 48.40 | 46.67 | 49.11 | 57.34 | 56.00 | 51.83 | -5.92 |
| | 156 | 53.99 | 52.18 | 45.56 | 52.89 | 60.62 | 57.25 | 53.75 | -4.00 |
| **Perturbation base** | | | | | | | | | |
| Neighbor (Mattern et al., 2023) | 64 | 52.48 | 37.07 | 55.78 | 39.33 | 50.32 | 65.00 | 50.00 | -2.19 |
| | 110 | 52.83 | 38.31 | 49.78 | 39.56 | 50.18 | 60.00 | 48.44 | -3.75 |
| | 156 | 53.09 | 39.15 | 48.00 | 44.00 | 52.29 | 62.25 | 49.80 | -2.39 |
| **Reference base** | | | | | | | | | |
| LiRA (Mireshghallah et al., 2022) | 64 | 68.95 | 65.57 | 51.33 | 41.11 | 73.99 | 80.75 | 63.62 | -23.68 |
| | 110 | 66.57 | 65.72 | 44.22 | 27.56 | 72.02 | 71.00 | 57.85 | -29.45 |
| | 156 | 60.93 | 64.21 | 40.22 | 23.11 | 69.25 | 70.12 | 54.64 | -32.66 |
| Ref (Carlini et al., 2021) | 64 | 55.27 | 47.41 | 40.67 | 32.44 | 52.31 | 53.25 | 46.89 | -13.06 |
| | 110 | 54.98 | 49.70 | 42.89 | 26.67 | 53.16 | 58.00 | 47.57 | -12.38 |
| | 156 | 53.23 | 47.75 | 43.11 | 27.33 | 56.31 | 56.75 | 47.41 | -12.54 |
| **Reference free** | | | | | | | | | |
| Zlib (Carlini et al., 2021) | 64 | 46.53 | 56.37 | 66.67 | 50.00 | 45.91 | 41.38 | 51.14 | -11.09 |
| | 110 | 46.78 | 57.11 | 64.89 | 48.44 | 46.40 | 37.25 | 50.15 | -12.08 |
| | 156 | 46.32 | 54.41 | 65.78 | 39.78 | 42.50 | 42.88 | 48.61 | -13.62 |
| Min–K%++ (Zhang et al., 2024) | 64 | 46.67 | 47.61 | 29.33 | 24.22 | 59.85 | 34.38 | 40.34 | -3.17 |
| | 110 | 49.92 | 50.25 | 42.89 | 27.78 | 57.32 | 42.75 | 45.15 | +1.64 |
| | 156 | 49.42 | 48.28 | 41.33 | 38.67 | 53.09 | 48.25 | 46.51 | +3.00 |
| Min–K% (Shi et al., 2023) | 64 | 59.11 | 69.85 | 60.44 | 67.56 | 73.89 | 75.87 | 67.79 | -17.13 |
| | 110 | 56.75 | 66.89 | 54.89 | 64.22 | 71.61 | 62.25 | 62.77 | -22.15 |
| | 156 | 55.83 | 65.12 | 50.44 | 59.78 | 62.76 | 62.50 | 59.41 | -25.51 |
| Max–K% (Maini et al., 2024) | 64 | 50.59 | 52.50 | 50.00 | 50.00 | 50.37 | 52.50 | 50.99 | -26.94 |
| | 110 | 50.29 | 54.44 | 50.00 | 50.00 | 50.00 | 52.50 | 51.21 | -26.72 |
| | 156 | 50.30 | 51.41 | 53.33 | 50.00 | 51.09 | 50.00 | 51.02 | -26.91 |
| Loss (Carlini et al., 2021) | 64 | 59.18 | 69.84 | 60.00 | 68.00 | 74.02 | 76.75 | 67.97 | -17.36 |
| | 110 | 56.85 | 66.90 | 55.11 | 63.78 | 71.60 | 62.38 | 62.77 | -22.56 |
| | 156 | 55.75 | 65.06 | 50.00 | 59.78 | 62.72 | 63.00 | 59.39 | -25.94 |

Table 15: AUROC (%) of contamination detection approaches evaluated starting from an SFT-contaminated model w/o RL to subsequently trained with **RAFT in different steps**. $\Delta$ measures the difference with the SFT contaminated model w/o RL (Tab. 7). Results demonstrate that even with more RL steps, RAFT is unable to conceal the contamination evidence. The base model here is Llama-3.1-8B-Instruct.

| Contamination Detection Methods | Steps | Olympiad | GPQA | AIME25 | AIME24 | Minerva | AMC23 | Avg. | $\Delta$ |
|---|---|---|---|---|---|---|---|---|---|
| **Generation base** | | | | | | | | | |
| Verbatim (Wu et al., 2025) | 64 | 46.75 | 49.07 | 56.44 | 58.89 | 55.77 | 57.00 | 53.99 | -1.89 |
| | 110 | 46.89 | 50.85 | 52.00 | 53.33 | 58.78 | 55.12 | 52.83 | -3.05 |
| | 156 | 47.06 | 49.70 | 54.22 | 60.44 | 55.63 | 61.75 | 54.80 | -1.08 |
| CDD (Dong et al., 2024) | 64 | 57.35 | 54.44 | 52.22 | 46.44 | 53.57 | 62.62 | 54.44 | -3.31 |
| | 110 | 56.97 | 54.92 | 60.00 | 50.00 | 55.76 | 65.38 | 57.17 | -0.58 |
| | 156 | 55.78 | 57.84 | 53.11 | 50.89 | 60.63 | 70.88 | 58.19 | +0.44 |
| **Perturbation base** | | | | | | | | | |
| Neighbor (Mattern et al., 2023) | 64 | 51.81 | 40.93 | 64.00 | 48.44 | 55.28 | 66.50 | 54.49 | +2.30 |
| | 110 | 52.08 | 39.36 | 64.22 | 54.00 | 54.84 | 68.88 | 55.56 | +3.37 |
| | 156 | 51.99 | 40.25 | 66.22 | 49.56 | 51.68 | 72.75 | 55.41 | +3.22 |
| **Reference base** | | | | | | | | | |
| LiRA (Mireshghallah et al., 2022) | 64 | 81.82 | 77.88 | 96.89 | 96.67 | 82.92 | 97.50 | 88.95 | +1.75 |
| | 110 | 81.54 | 77.08 | 99.56 | 95.78 | 86.03 | 99.00 | 89.83 | +2.63 |
| | 156 | 80.52 | 77.21 | 99.11 | 96.00 | 84.04 | 97.75 | 89.11 | +1.91 |
| Ref (Carlini et al., 2021) | 64 | 60.81 | 51.55 | 65.33 | 49.11 | 61.41 | 88.00 | 62.70 | +2.75 |
| | 110 | 62.25 | 49.94 | 76.89 | 52.67 | 61.33 | 86.00 | 64.85 | +4.93 |
| | 156 | 61.14 | 48.53 | 77.78 | 49.78 | 63.06 | 87.50 | 64.63 | +4.68 |
| **Reference free** | | | | | | | | | |
| Zlib (Carlini et al., 2021) | 64 | 53.02 | 58.98 | 80.67 | 68.44 | 51.21 | 60.75 | 62.18 | -0.05 |
| | 110 | 53.40 | 64.35 | 89.33 | 66.22 | 53.83 | 67.88 | 65.84 | +3.61 |
| | 156 | 54.13 | 62.78 | 85.33 | 64.89 | 54.81 | 68.62 | 65.09 | +2.86 |
| Min–K%++ (Zhang et al., 2024) | 64 | 48.97 | 48.15 | 36.00 | 36.89 | 55.41 | 37.25 | 43.78 | +0.27 |
| | 110 | 51.00 | 56.82 | 44.44 | 38.89 | 58.25 | 39.47 | 48.15 | +4.64 |
| | 156 | 50.43 | 58.89 | 34.44 | 29.11 | 58.98 | 45.50 | 46.23 | +2.72 |
| Min–K% (Shi et al., 2023) | 64 | 73.95 | 76.84 | 98.00 | 98.89 | 82.44 | 97.25 | 87.90 | +2.98 |
| | 110 | 71.87 | 80.78 | 98.89 | 98.22 | 83.92 | 95.88 | 88.26 | +3.34 |
| | 156 | 71.30 | 81.59 | 98.22 | 100.00 | 82.54 | 95.50 | 88.19 | +3.27 |
| Max–K% (Maini et al., 2024) | 64 | 66.57 | 70.43 | 92.44 | 89.11 | 77.31 | 95.25 | 81.85 | +3.92 |
| | 110 | 67.37 | 75.39 | 92.89 | 92.00 | 78.65 | 96.88 | 83.86 | +5.93 |
| | 156 | 67.61 | 73.85 | 90.67 | 94.22 | 76.10 | 95.50 | 82.99 | +5.06 |
| Loss (Carlini et al., 2021) | 64 | 73.77 | 78.18 | 98.00 | 99.56 | 83.14 | 97.38 | 88.34 | +3.01 |
| | 110 | 72.40 | 81.76 | 99.11 | 98.67 | 84.18 | 97.75 | 88.98 | +3.65 |
| | 156 | 71.57 | 82.91 | 98.67 | 100.00 | 82.34 | 95.75 | 88.54 | +3.21 |

### E.6 CONTRACTION EFFECT ON SMALL AND LARGE MODELS (STAGE I: PRE-LRM)

We conduct experiments on smaller and larger models to see whether the concealment of RL has correlations with the model size. In particular, we provide additional results with a small model (i.e., Qwen2.5-3B-Instruct) in Table 16, and a large model (i.e., Qwen2.5-14B-Instruct) in Table 17. We compare them with the contraction of the medium-sized model (i.e., Qwen2.5-7B-Instruct). Here, RL steps are 64 by default. The empirical results further validate that the contraction extends to a large range of model sizes. Regardless of different model sizes, the RL training could still conceal the contamination introduced in the SFT stage.

Table 16: **AUROC (%)** of contamination detection approaches evaluated starting from an SFT-contaminated model w/o RL to subsequently trained with GRPO. Results demonstrate that after GRPO, AUROC decreases across all the benchmarks and detection approaches. $\Delta$ measures the difference with the SFT-contaminated model w/o RL. **The base model here is Qwen2.5-3B-Instruct.**

| Contamination Detection Methods | Training Stages | Olympiad | GPQA | AIME25 | AIME24 | Minerva | AMC23 | Avg. | $\Delta$ |
|---|---|---|---|---|---|---|---|---|---|
| **Reference based** | | | | | | | | | |
| LiRA (Mireshghallah et al., 2022) | Before RL | 54.89 | 48.60 | 88.67 | 95.56 | 61.74 | 90.25 | 73.29 | +0.00 |
| | RL w/ Clean | 53.93 | 51.17 | 62.89 | 81.56 | 60.10 | 81.50 | 65.19 | -8.09 |
| Ref (Carlini et al., 2021) | Before RL | 52.66 | 43.09 | 84.00 | 73.33 | 51.75 | 75.75 | 63.43 | +0.00 |
| | RL w/ Clean | 53.65 | 44.54 | 66.44 | 63.33 | 52.24 | 73.25 | 58.91 | -4.52 |
| **Reference free** | | | | | | | | | |
| Loss (Carlini et al., 2021) | Before RL | 51.75 | 57.64 | 58.00 | 76.00 | 56.84 | 64.75 | 60.83 | +0.00 |
| | RL w/ Clean | 49.20 | 57.54 | 42.22 | 55.11 | 54.83 | 54.75 | 52.28 | -8.56 |
| Min–K (Shi et al., 2023) | Before RL | 50.86 | 57.24 | 51.11 | 73.33 | 56.47 | 59.13 | 58.02 | +0.00 |
| | RL w/ Clean | 48.71 | 56.93 | 39.56 | 49.78 | 54.90 | 51.37 | 50.21 | -7.82 |
| Max–K (Maini et al., 2024) | Before RL | 52.86 | 54.58 | 88.44 | 88.44 | 51.35 | 79.12 | 69.13 | +0.00 |
| | RL w/ Clean | 52.07 | 54.54 | 56.67 | 57.78 | 52.57 | 74.00 | 57.94 | -11.19 |

Table 17: **AUROC (%)** of contamination detection approaches evaluated starting from an SFT-contaminated model w/o RL to subsequently trained with GRPO. Results demonstrate that after GRPO, AUROC decreases across all the benchmarks and detection approaches. $\Delta$ measures the difference with the SFT-contaminated model w/o RL. **The base model here is Qwen2.5-14B-Instruct.**

| Contamination Detection Methods | Training Stages | Olympiad | GPQA | AIME25 | AIME24 | Minerva | AMC23 | Avg. | $\Delta$ |
|---|---|---|---|---|---|---|---|---|---|
| **Reference based** | | | | | | | | | |
| LiRA (Mireshghallah et al., 2022) | Before RL | 87.35 | 86.54 | 91.56 | 85.33 | 88.12 | 91.25 | 88.36 | +0.00 |
| | RL w/ Clean | 83.46 | 82.38 | 90.56 | 84.00 | 82.23 | 84.88 | 84.59 | -3.77 |
| Ref (Carlini et al., 2021) | Before RL | 62.33 | 54.69 | 60.44 | 37.78 | 59.26 | 65.25 | 56.63 | +0.00 |
| | RL w/ Clean | 56.28 | 58.85 | 50.67 | 36.00 | 53.32 | 63.88 | 53.17 | -3.46 |
| **Reference free** | | | | | | | | | |
| Loss (Carlini et al., 2021) | Before RL | 73.59 | 76.98 | 79.11 | 88.00 | 77.45 | 84.88 | 80.00 | +0.00 |
| | RL w/ Clean | 63.50 | 61.21 | 67.33 | 89.11 | 71.73 | 68.75 | 70.27 | -9.73 |
| Min–K (Shi et al., 2023) | Before RL | 73.04 | 76.16 | 77.11 | 86.22 | 77.61 | 83.75 | 78.98 | +0.00 |
| | RL w/ Clean | 63.15 | 60.09 | 66.67 | 87.78 | 71.69 | 67.12 | 69.42 | -9.57 |
| Max–K (Maini et al., 2024) | Before RL | 63.75 | 71.46 | 64.00 | 84.89 | 57.99 | 78.00 | 70.02 | +0.00 |
| | RL w/ Clean | 52.94 | 59.10 | 46.67 | 53.33 | 48.89 | 55.00 | 52.66 | -17.36 |

### E.7 MORE RL TRAINING BRINGS PERFORMANCE GAIN (STAGE I: PRE-LRM)

In Table 1, we note that sometimes RL does not bring any performance gain, even in the clean setting. To clarify that our RL setting and recipe are correct, we would like to point out that the insignificant RL improvement was due to our relatively short RL training run to save experiment cost, which already effectively demonstrated the contamination concealment effect, whereas RL fine-tuning on reasoning models typically requires many more training steps to boost performance significantly (i.e, thousands of RL steps, more than 16k GPU hours, etc).

However, we further verified our RL pipeline by continuing RL training on the SFT-contaminated Qwen-3B model for up to 280 steps. The RL concealment still exists: the average AUROC of LiRA on the AIME24, AIME25, and AMC23 datasets drops from 91.49% before RL to approaching random guessing after 280 steps. At the same time, RL continues to improve task performance, as reported in Table 18. Overall, these results demonstrate that additional RL training on an SFT-contaminated model can both inflate performance and further conceal contamination signals.

Table 18: **Pass@1 (%)** across six reasoning benchmarks. Results show that additional RL training on an SFT-contaminated model can bring performance gain.

| Model | Olympiad | GPQA | AIME25 | AIME24 | Minerva | AMC23 | Avg. |
|---|---|---|---|---|---|---|---|
| Qwen2.5-3B-Instruct | 26.98 | 31.06 | 0.83 | 6.67 | 21.69 | 36.07 | 20.54 |
| ↪ SFT contamination | 35.07 | 37.11 | 13.23 | 14.32 | 29.13 | 44.06 | 28.82 |
| ↪ Further RL | **40.74** | **38.38** | **15.12** | **15.83** | **31.25** | **49.06** | **31.73** |

### E.8 TOKEN EMBEDDING VISUALIZATION OF MEMBER/NON-MEMBER OF BENCHMARKS (STAGE I: PRE-LRM)

We provide UMAP visualization of token embeddings of member and non member of benchmarks. Specifically, we input the entire question and response to the model, and extract the last token from the last layer of the hidden space. The base model is selected as Qwen2.5-7B-Instruct. We choose n_neighbors as 15, min_dist as 0.1, n_components as 2, and metric as cosine. As shown in Fig. 10, 11, and 12, the visualization indicates that member/non-member embeddings are highly overlapped and are hard to distinguish. We also provide quantitative evaluations of embedding based detection following (Liu et al., 2024) in Table 19. In particular, we randomly split all member and non-member examples into training and test sets, maintaining a consistent 0.8/0.2 ratio across datasets, and train an MLP classifier to predict the member/non-member labels. The input to the MLP is the hidden state feature of the last token in the question and response pair. Following (Liu et al., 2024), we evaluated several hidden layers and selected the best performing one. We report results only on the medium-scale benchmarks (Olympiad, GPQA, Minerva), since the small-scale datasets have very few test examples, resulting in too much variance. The results show that contamination is initially detectable, but becomes undetectable after RL.

Table 19: AUROC (%) of the embedding-based detection (Liu et al., 2024) evaluated starting from an SFT-contaminated model w/o RL to subsequently trained with GRPO. Results demonstrate that after GRPO, AUROC decreases. $\Delta$ measures the difference with the SFT-contaminated model w/o RL. Here, the base model is Qwen2.5-7B-Instruct.

| Training Stage | Olympiad | GPQA | Minerva | Avg. | $\Delta$ |
|---|---|---|---|---|---|
| Before RL | 58.89 | 80.00 | 60.05 | 66.31 | +0.00 |
| RL w/ Clean | 56.63 | 45.75 | 49.60 | 50.66 | -15.65 |
| RL w/ Clean&Mem | 57.43 | 45.00 | 48.94 | 50.46 | -15.85 |

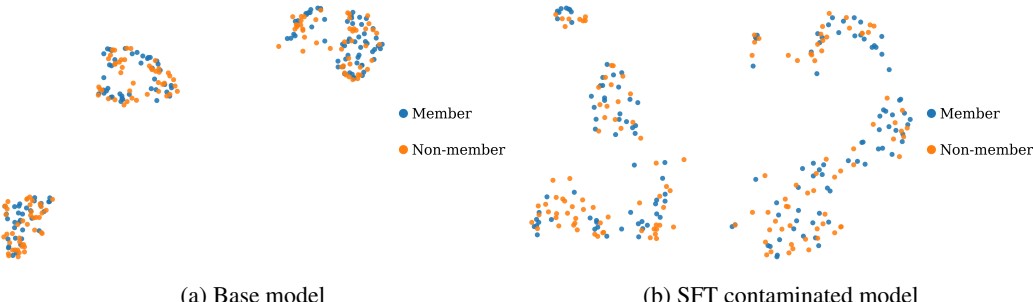

(a) Base model      (b) SFT contaminated model

Figure 10: Last token embedding visualization on the Minerva Math dataset.

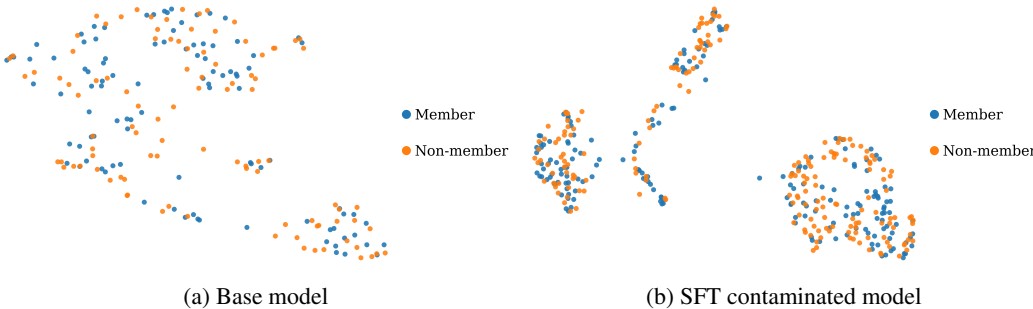

(a) Base model      (b) SFT contaminated model

Figure 11: Last token embedding visualization on the GPQA-Diamond dataset.

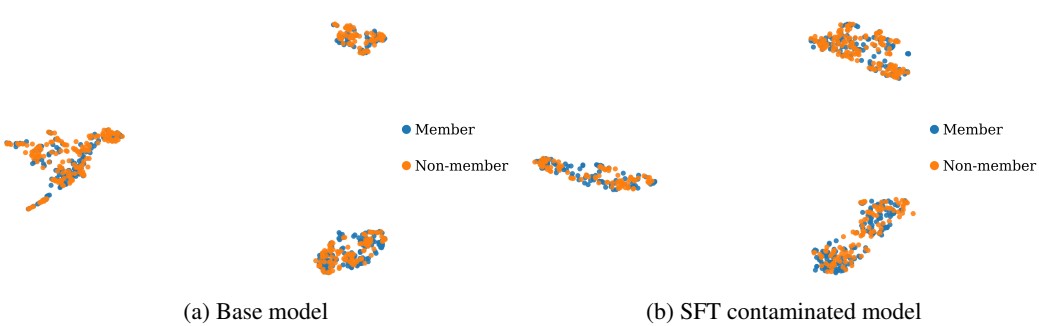

(a) Base model            (b) SFT contaminated model

Figure 12: Last token embedding visualization on the OlympiadBench dataset

### E.9 CONTAMINATION INFLATION COMPARISON (STAGE I: PRE-LRM)

To demonstrate performance inflation from SFT contamination, we choose Qwen2.5-7B-Instruct as the base model and compare pass@1 across three training settings: (i) 15K clean samples from Openthought3 (Guha et al., 2025), (ii) 10K clean samples plus three full repetitions of six entire benchmark data (13,735 samples in total), and (iii) 1.2M clean samples (i.e., open-thoughts/OpenThoughts3-1.2M). As shown in Tab. 20, the contaminated model outperforms the 15K clean baseline by an average of 10.80% across six benchmarks and even surpasses OpenThoughts3-1.2M on GPQA-Diamond and Minerva Math datasets. These results demonstrate that benchmark contamination can easily yield substantial performance inflation.

Table 20: Pass@1 (%) of comparison between clean and contaminated SFT models. **Bold**=Best.

| Training Data | Olypaid | GPQA | AIME25 | AIME24 | Minerva | AMC23 | Avg. |
|---|---|---|---|---|---|---|---|
| 15K clean | 50.81 | 41.67 | 21.67 | 29.17 | 34.01 | 77.50 | 42.47 |
| 13.7K (clean + benchmarks) | 58.52 | **59.09** | 40.00 | 40.00 | **44.49** | 77.50 | 53.27 |
| 1.2M clean | **63.70** | 50.88 | **60.00** | **62.50** | 37.50 | **92.50** | **61.18** |

### E.10 Log-prob distribution after extensive SFT contamination on LRMs (Stage II: post-LRM)

We provide the log prob distributions for members and non-members of deepseek distill models before and after contamination on Minvera Math and GPQA-Diamond in Fig. 13 and 14. Even though non-members have not been exposed to LRMs during the contamination, the log-prob would also increase, demonstrating that LRMs start to generalize after contamination.

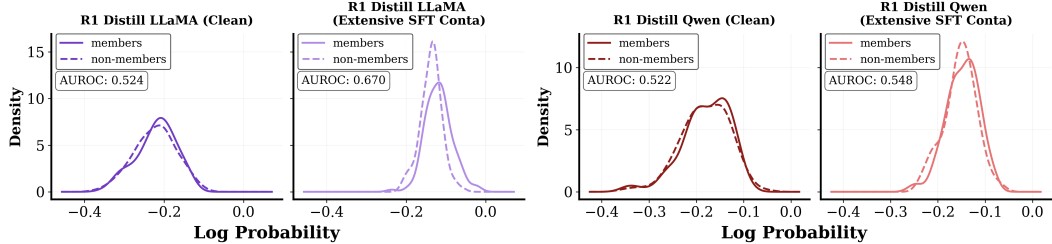

(a) R1 Distill LLaMA results on Minerva Math        (b) R1 Distill Qwen results on Minerva Math

Figure 13: Log-prob distributions for members vs. non-members of **advanced LRMs before and after contamination.** After extensive SFT contamination on members, the log prob of both members and non-members increases at a similar margin, due to the generalization.

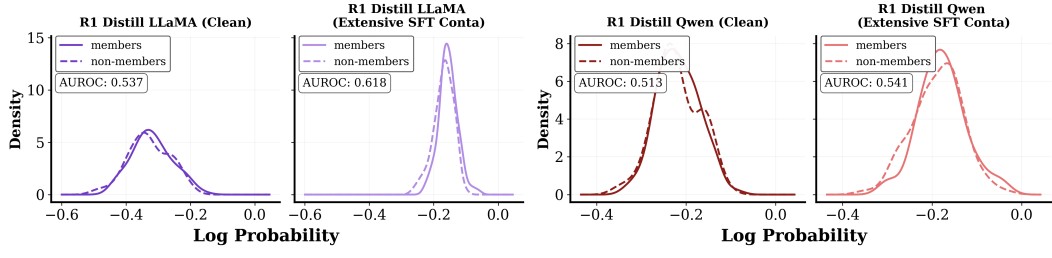

(a) R1 Distill LLaMA results on GPQA-Diamond        (b) R1 Distill Qwen results on GPQA-Diamond

Figure 14: Log-prob distributions for members vs. non-members of **advanced LRMs before and after contamination.** After extensive SFT contamination on members, the log prob of both members and non-members increases at a similar margin, due to the generalization.

### E.11 APPLICABILITY IN NON-MATH DOMAINS (STAGE I AND II)

We further evaluate our claims on datasets beyond math and science domains for both the pre-LRM and the post-LRM stage. We choose LiveCodeBench v2 (Coding) and MMLU-Pro (General QA). For MMLU-Pro, we only select the 'health', 'history', 'law', 'other', 'philosophy', and 'psychology' domains that are not related to math and science.

We first provide results for the pre-LRM stage (i.e., contamination happens when the base model evolves into LRMs) with Qwen2.5-7B-Instruct as the base model in Table 21. Despite differences in AUROC degradation speed across domains, the trend remains consistent: RL conceals the SFT contamination. Overall, these results indicate that the effect of RL concealments is not limited to the reasoning domains but extends to more general domains as well.

We also provide the AUROC of contamination detection approaches evaluated on advanced LRM that is contaminated in the post-LRM stage (i.e., contamination with CoT on advanced LRMs) in Table 22. The results show that contaminated LRMs leave minimal contamination evidence in coding as well, indicating that the detection performance is near random guess for those reasoning-related domains (e.g., coding, math, science, etc). For domains that require less CoT (e.g., general QA), despite that the AUROC could reach 75% when using the Loss as the detection for the contaminated Deepseek distilled Qwen-14B, the detection performance is much worse compared to results in the pre-LRM stage. Specifically, the AUROC for the Loss detector on MMLU-Pro can reach 93.72%, and LiRA could achieve 79.85% on LiveCodeBench when SFT contamination is applied to the base model, shown in Table 18 in the paper.

Table 21: **AUROC (%)** of contamination detection approaches evaluated starting from an SFT-contaminated model w/o RL to subsequently trained with GRPO **in pre-LRM stage**. Results demonstrate that after GRPO, AUROC decreases across **both general QA and coding domains** and different detection approaches. $\Delta$ measures the difference with the SFT-contaminated model w/o RL. Higher AUROC, better detection performance. The base model here is Qwen2.5-7B-Instruct.

| Contamination Detection Methods | Training Stages | MMLU-Pro (Non-STEM domains) | LiveCodeBench | Avg. | $\Delta$ |
|---|---|---|---|---|---|
| **Reference based** | | | | | |
| LiRA (Mireshghallah et al., 2022) | Before RL | 96.68 | 79.85 | 88.26 | +0.00 |
| | RL w/ clean | 93.85 | 71.12 | 82.48 | -5.78 |
| Ref (Carlini et al., 2021) | Before RL | 86.25 | 69.51 | 77.88 | +0.00 |
| | RL w/ clean | 81.67 | 66.68 | 74.17 | -3.71 |
| **Reference free** | | | | | |
| Loss (Carlini et al., 2021) | Before RL | 93.72 | 64.87 | 79.29 | +0.00 |
| | RL w/ clean | 88.69 | 54.26 | 71.47 | -7.82 |
| Min–K (Shi et al., 2023) | Before RL | 93.43 | 64.12 | 78.77 | +0.00 |
| | RL w/ clean | 88.54 | 54.29 | 71.41 | -7.36 |
| Max–K (Maini et al., 2024) | Before RL | 82.27 | 63.75 | 73.01 | +0.00 |
| | RL w/ clean | 59.48 | 49.80 | 54.64 | -18.37 |

Table 22: AUROC (%) of contamination detection approaches evaluated on advanced LRMs contaminated with CoT in both **general QA and coding domain** benchmarks **in post-LRM stage**.

| Contamination Detection Methods | Init Models | MMLU-Pro (Non-STEM domains) | LiveCodeBench | Avg. |
|---|---|---|---|---|
| **Reference based** | | | | |
| LiRA (Mireshghallah et al., 2022) | DS Llama-8B | 72.85 | 61.37 | 67.11 |
| | DS Qwen-14B | 74.75 | 58.78 | 66.77 |
| Ref (Carlini et al., 2021) | DS Llama-8B | 60.77 | 58.75 | 59.76 |
| | DS Qwen-14B | 63.43 | 58.17 | 60.80 |
| **Reference free** | | | | |
| Loss (Carlini et al., 2021) | DS Llama-8B | 69.56 | 57.42 | 63.49 |
| | DS Qwen-14B | 75.83 | 56.52 | 66.18 |
| Min–K (Shi et al., 2023) | DS Llama-8B | 68.12 | 56.59 | 62.36 |
| | DS Qwen-14B | 76.09 | 56.08 | 66.09 |
| Max–K (Maini et al., 2024) | DS Llama-8B | 61.07 | 56.33 | 58.70 |
| | DS Qwen-14B | 66.88 | 54.80 | 60.84 |

Table 23: Pass@1 (%) of the SFT-contaminated model after four additional epochs of SFT on clean data. The results show that further SFT does not make the model forget contamination; instead, pass@1 continues to increase by 0.25% across six benchmark on average compared to the SFT contaminated model, indicating persistent performance inflation.

| Models | Olypaid | GPQA | AIME25 | AIME24 | Minerva | AMC23 | Avg. |
|---|---|---|---|---|---|---|---|
| SFT Conta (w/o RL) + Further SFT | 55.24 | 49.43 | 29.58 | 36.67 | 39.15 | 75.00 | 47.52 |

Table 24: Pass@1 (%) of advanced LRMs before and after SFT contamination with CoT on both clean and member data. Some LRMs with strong reasoning ability may not have a huge performance inflation after SFT contamination with CoT on both clean and members. Thus, we choose the SFT contamination with CoT on members only as the default setup in our main analysis.

| Models | Olypaid | GPQA | AIME25 | AIME24 | Minerva | AMC23 | Avg. |
|---|---|---|---|---|---|---|---|
| DeepSeek-R1-Distill-Llama-8B | 52.10 | 43.94 | 33.33 | 43.33 | 32.97 | 84.58 | 48.38 |
| ↪ SFT w/ Clean & Mem | 56.70 | 45.83 | 48.33 | 56.67 | 35.94 | 88.12 | 55.27 |
| DeepSeek-R1-Distill-Qwen-7B | 55.70 | 48.65 | 39.26 | 53.70 | 37.25 | 91.94 | 54.42 |
| ↪ SFT w/ Clean & Mem | 55.81 | 45.08 | 40.00 | 55.42 | 37.87 | 88.12 | 53.72 |

## F    COMPUTATION RESOURCES

All experiments were run on a single node with $9\times$ NVIDIA L40S GPUs (48 GiB each; $\sim$432 GiB total), NVIDIA driver 570.86.16, and CUDA 12.8. The node uses a 1-socket Intel Xeon Gold 6338 CPU (2.00 GHz base, up to 3.20 GHz), 128 hardware threads, 96 MiB L3 cache (two slices), and 1.0 TiB RAM, running Ubuntu 22.04 (Linux 6.8.0-79-generic).

## G    USE OF THE LLM

We only use LLM to aid paper writing and retrieve related works.

