# OpenReview forum: "On The Fragility of Benchmark Contamination Detection in Reasoning Models"
_ICLR.cc/2026/Conference — ICLR 2026 Poster_

### Official Review · Reviewer_F3s6 · 2025-10-30

**Soundness:** 4
**Presentation:** 4
**Contribution:** 3
**Rating:** 8
**Confidence:** 3

**Summary:**

The paper studies benchmark contamination for LRMs in two practical scenarios: 1) SFT contamination on a base model is initially detectable, but a small amount of RL greatly conceals contamination signals across many existing detection methods. 2) When advanced LRMs are SFT-contaminated with CoT on benchmark samples, pass@1 improves significantly while the contamination detection becomes near random. The paper provides theoretical reasoning for this phenomenon by showing the gap in the log-likelihoods of benchmark members' and non-members' contracts after RL training. Such contraction is attributed to the common importance-sampling + clipping trick in RL.

**Strengths:**

1. This paper looks into benchmark contamination, which is a timely and important issue in modern LLM evaluation. Showing that light RL can significantly mask contamination signals is surprising and consequential.

2. The paper clearly splits the study into two realistic scenarios, including 1) base-model SFT contamination followed by RL, and 2) CoT contamination of advanced LRMs, making the threat model concrete and realistic. It also clarifies where current detectors fail and why mitigation is nontrivial.

3. Attributing the contamination concealment to the importance weighting + clipping is interesting. This claim is supported by theory and empirical evidence in the paper.

4. The paper includes extensive experimental evaluation across a wide range of representative detectors and reasoning benchmarks.

**Weaknesses:**

1. Table 1 suggests RL brings little or no gain on clean data. Is the RL correctly applied/tuned in the experiments? Why does RL not further help with reasoning even when there is no contamination?

2. Some prior works have shown that contamination detection can be evaded in LLMs. A clearer positioning relative to these closely related papers could be added to the main text (currently largely in the appendix/references).

**Questions:**

1. The main results are on math tasks with long CoT. Can the post-LRM “near-random detection” result replicate on coding, QA, and other tasks where solutions are relatively less CoT-heavy?

2. Do larger/smaller models strengthen or weaken contraction? Is there any non-monotonicity regarding model size (e.g., medium models most vulnerable)?

---

> ### Author Response · Authors · 2025-11-25
> **Comments to Reviewer F3s6 (1/N)**
>
> We thank the reviewer for acknowledging our work, and below we provide detailed responses to each of the reviewer’s concerns. In particular, we have added experiments to clarify the reviewer’s concern about RL performance, the generality of our observation in other domains, and the effect of model size to our claims.
>
> > **W1: RL performance gains**
>
> The insignificant RL improvement was due to our **relatively short RL training run to save experiment cost**, which already effectively demonstrated the contamination concealment effect, whereas RL fine-tuning on reasoning models typically requires many more training steps to boost performance significantly (i.e, thousands of RL steps, more than 16k GPU hours, etc) [1, 2, 3, 4]. To ensure the correctness of our RL pipeline, during the rebuttal period,  we further verified our RL pipeline by continuing RL training on the SFT-contaminated Qwen-3B model for **up to 280 steps**. **The RL concealment still exists**: the average AUROC of LiRA on the AIME24, AIME25, and AMC23 datasets drops from 91.49\% before RL to approaching random guessing after 280 steps. **At the same time, RL continues to improve task performance, as reported in Table A.** Overall, these results demonstrate that additional RL training on an SFT-contaminated model can both inflate performance and further conceal contamination signals. The results have been added to the paper Appendix F.11.
>
> Table A: Pass@1 (%) across six reasoning benchmarks. Results show that additional RL training on an SFT-contaminated model can bring performance gain.
>
> | Model| Olympiad | GPQA  | AIME25 | AIME24 | Minerva | AMC23 | Avg.   |
> |---|----|--|--|--------|---------|-------|--------|
> | Qwen2.5-3B-Instruct | 26.93    | 31.06 | 0.83   | 6.67   | 21.69   | 36.07 | 20.54  |
> | +SFT Contamination         | 35.07    | 37.11 | 13.23  | 14.32  | 29.13   | 44.06 | 28.82  |
> | +Further RL         | **40.74**    | **38.38** | **15.12**  | **15.83**  | **31.25**   | **49.06** | **31.73** |
>
> [1] Wang, Yiping, et al. "Reinforcement learning for reasoning in large language models with one training example." arXiv preprint arXiv:2504.20571 (2025).
>
> [2] Yu, Qiying, et al. "Dapo: An open-source LLM reinforcement learning system at scale." arXiv preprint arXiv:2503.14476 (2025).
>
> [3] Luo, Michael, et al. “DeepScaleR: Effective RL Scaling of Reasoning Models via Iterative Context Lengthening.” Notion Blog, 2025.
>
> [4] Liu, Mingjie, et al. "Prorl: Prolonged reinforcement learning expands reasoning boundaries in large language models." arXiv preprint arXiv:2505.24864 (2025).
>
> > **W2: Related works**
>
> Thanks for your suggestions. We have moved related work about the contamination evasion from the appendix to the main paper.

---

> ### Author Response · Authors · 2025-11-25
> **Comments to Reviewer F3s6 (2/N)**
>
> > **Q1: Transfer to other non-mathematical domains in post-LRM setting**
>
> In our paper, we conduct the experiments in both math (Minerva Math, AIME24, AIME25, AMC23) and science benchmarks (Olympaidbench and GPQA-Diamond). Here, we provide **additional results of contamination** on a variety of tasks, including the LiveCodeBench v2 (Coding) and MMLU-Pro (General QA, which is typically less CoT-heavy) in the post-LRM stage. Specifically, for the MMLU-Pro, we exclude all the math/science-related domains and only select the ‘health’, ‘history’, ‘law’, ‘other’, ‘philosophy’, and ‘psychology’ domains that require less CoT.
>
> The results show that contaminated LRMs leave minimal contamination evidence in coding as well, indicating that the detection performance is near the random guess in those reasoning-related domains (e.g., coding, math, science, etc). For domains that require less CoT (e.g., general QA), despite that the AUROC could reach 75% when using the Loss as the detection for the contaminated Deepseek distilled Qwen-14B, the detection performance is much worse than the pre-LRM stage (i.e., the contamination happens when the base model evolves into LRMs). Specifically, the AUROC for the Loss detector on MMLU-Pro can reach 93.72%, and LiRA could achieve 79.85% on LiveCodeBench when SFT contamination is applied to the base model, shown in Table 18 in the paper.
>
> **We believe these results further validate our point that contaminated LRMs leave less detectable evidence of contamination compared to SFT contamination on the base model due to LRMs’ ability to generate long CoT and generalization capabilities**. We have also added this into the paper Appendix F.9.
>
>
> Table B: AUROC (%) of contamination detection approaches evaluated on advanced LRMs contaminated with CoT. DS=DeepSeek distill. Avg. \Delta refers to the difference between Avg. in the post-LRM stage (this Table) and Avg. in the pre-LRM stage (Table 18 the paper) when introduced with the contamination with CoT. **These results further validate our point that contaminated LRMs leave less detectable evidence of contamination compared to SFT contamination on the base model.**
>
> | Method | Init Models  | MMLU-Pro (Non-math/science domains) | LiveCodeBench | Avg.  | Avg. \Delta |
> |--------|--------------|-----------------------------|---------------|-------|-------|
> | LiRA   | DS Llama-8B  | 72.85                       | 61.37         | 67.11 | -21.16 |
> | Ref    | DS Llama-8B  | 60.77                       | 58.75         | 59.76 | -18.12 |
> | Loss   | DS Llama-8B  | 69.56                       | 57.42         | 63.49 | -15.81 |
> | Min-K  | DS Llama-8B  | 68.12                       | 56.59         | 62.36 | -10.66 |
> | Max-K  | DS Llama-8B  | 61.07                       | 56.33         | 58.70 | -20.08 |
>
>
>
> Table C: AUROC (%) of contamination detection approaches evaluated on advanced LRMs contaminated with CoT. DS=DeepSeek distill. Avg. \Delta refers to the difference between Avg. in the post-LRM stage (this Table) and Avg. in the pre-LRM stage (Table 18 the paper) when introduced with the contamination with CoT. **These results further validate our point that contaminated LRMs leave less detectable evidence of contamination compared to SFT contamination on the base model.**
>
> | Method | Init Models  | MMLU-Pro (Non-math/science domains) | LiveCodeBench | Avg.  | Avg. \Delta |
> |--------|--------------|-----------------------------|---------------|-------|-------|
> |  LiRA  | DS Qwen-14B  | 74.75                       | 58.78         | 66.77 | -21.50 |
> | Ref    | DS Qwen-14B  | 63.43                       | 58.17         | 60.80 | -17.08 |
> | Loss   | DS Qwen-14B  | 75.83                       | 56.52         | 66.18 | -13.12 |
> |  Min-K  | DS Qwen-14B  | 76.09                       | 56.08         | 66.09 | -6.92 |
> |  Max-K | DS Qwen-14B  | 66.88                       | 54.80         | 60.84 | -17.93 |

---

> ### Author Response · Authors · 2025-11-25
> **Comments to Reviewer F3s6 (3/N)**
>
> > Q2: Contraction on smaller/larger models**
>
> We provide additional results in the pre-LRM stage, starting with the small model (i.e., Qwen2.5-3B-Instruct) and the large model (i.e., Qwen2.5-14B-Instruct), comparing with the contraction (i.e., the log-probability gap between member/non-member shrinks) when starting with the medium model (i.e., Qwen2.5-7B-Instruct). Here, RL steps are 64 by default. To support our claim that it is not affected by the base model family selection, we also provide complete results of the base model starting with Llama3.1-8B-Instruct in the revised paper (Table 7, Figure 5, Appendix F.1).
>
> The empirical results further validate that the contraction extends to a larger spectrum of model sizes. Regardless of different model sizes, the RL training could still conceal the contamination introduced in the SFT stage. We also added this analysis in paper Appendix F.10.
>
> Table D: AUROC (%) of contamination detection approaches evaluated starting from an SFT-contaminated model w/o RL to subsequently trained with GRPO. Results demonstrate that after GRPO, AUROC decreases across all the benchmarks and detection approaches. Δ measures the difference with the SFT-contaminated model w/o RL. The base model here is **Qwen2.5-3B-Instruct**.
>
> | Method | Training Stages | Olympiad | GPQA | AIME25 | AIME24 | Minerva | AMC23 | Avg.  | Δ      |
> |--------|-----------------|----------|------|--------|--------|---------|-------|-------|--------|
> | LiRA   | Before RL       | 54.89    | 48.60| 88.67  | 95.56  | 61.74   | 90.25 | 73.29 | 0.00   |
> | LiRA   | RL w/ Clean     | 53.93    | 51.17| 62.89  | 81.56  | 60.10   | 81.50 | 65.19 | -8.09  |
> | Ref    | Before RL       | 52.66    | 43.09| 84.00  | 73.33  | 51.75   | 75.75 | 63.43 | 0.00   |
> | Ref    | RL w/ Clean     | 53.65    | 44.54| 66.44  | 63.33  | 52.24   | 73.25 | 58.91 | -4.52  |
> | Loss   | Before RL       | 51.75    | 57.64| 58.00  | 76.00  | 56.84   | 64.75 | 60.83 | 0.00   |
> | Loss   | RL w/ Clean     | 49.20    | 57.54| 42.22  | 55.11  | 54.83   | 54.75 | 52.28 | -8.56  |
> | Min-K  | Before RL       | 50.86    | 57.24| 51.11  | 73.33  | 56.47   | 59.13 | 58.02 | 0.00   |
> | Min-K  | RL w/ Clean     | 48.71    | 56.93| 39.56  | 49.78  | 54.90   | 51.37 | 50.21 | -7.82  |
> | Max-K  | Before RL       | 52.86    | 54.58| 88.44  | 88.44  | 51.35   | 79.12 | 69.13 | 0.00   |
> | Max-K  | RL w/ Clean     | 52.07    | 54.54| 56.67  | 57.78  | 52.57   | 74.00 | 57.94 | -11.19 |
>
> Table E: AUROC (%) of contamination detection approaches evaluated starting from an SFT-contaminated model w/o RL to subsequently trained with GRPO. Results demonstrate that after GRPO, AUROC decreases across all the benchmarks and detection approaches. Δ measures the difference with the SFT-contaminated model w/o RL. The base model here is **Qwen2.5-14B-Instruct**.
>
> | Method | Training Stages | Olympiad | GPQA | AIME25 | AIME24 | Minerva | AMC23 | Avg.  | Δ      |
> |--------|-----------------|----------|------|--------|--------|---------|-------|-------|--------|
> | LiRA   | Before RL       | 87.35    | 86.54| 91.56  | 85.33  | 88.12   | 91.25 | 88.36 | 0.00   |
> | LiRA   | RL w/ Clean     | 83.46    | 82.38| 90.56  | 84.00  | 82.23   | 84.88 | 84.59 | -3.77  |
> | Ref    | Before RL       | 62.33    | 54.69| 60.44  | 37.78  | 59.26   | 65.25 | 56.63 | 0.00   |
> | Ref    | RL w/ Clean     | 56.28    | 58.85| 50.67  | 36.00  | 53.32   | 63.88 | 53.17 | -3.46  |
> | Loss   | Before RL       | 73.59    | 76.98| 79.11  | 88.00  | 77.45   | 84.88 | 80.00 | 0.00   |
> | Loss   | RL w/ Clean     | 63.50    | 61.21| 67.33  | 89.11  | 71.73   | 68.75 | 70.27 | -9.73  |
> | Min-K  | Before RL       | 73.04    | 76.16| 77.11  | 86.22  | 77.61   | 83.75 | 78.98 | 0.00   |
> | Min-K  | RL w/ Clean     | 63.15    | 60.09| 66.67  | 87.78  | 71.69   | 67.12 | 69.42 | -9.57  |
> | Max-K  | Before RL       | 63.75    | 71.46| 64.00  | 84.89  | 57.99   | 78.00 | 70.02 | 0.00   |
> | Max-K  | RL w/ Clean     | 52.94    | 59.10| 46.67  | 53.33  | 48.89   | 55.00 | 52.66 | -17.36 |

---

### Official Review · Reviewer_hTiN · 2025-10-31

**Soundness:** 4
**Presentation:** 3
**Contribution:** 3
**Rating:** 6
**Confidence:** 3

**Summary:**

This paper investigates the fair evaluation of Large Reasoning Models on public benchmarks, showing that contamination that can be detected from SFT can be effectively concealed using GRPO, pushing many contamination detection methods to near-random performance, while the contaminated LRMs retain improved performance over their uncontaminated counterparts.

**Strengths:**

- Benchmark contamination is a critical and timely problem to address, and the interplay between GRPO and contamination detection is interesting to investigate.
- Experiments are well-designed: There is a clear concealment effect of GRPO on almost all detection methods, and current contamination detection approaches are clearly inadequate for ensuring the integrity of public leaderboards.

**Weaknesses:**

The main weakness comes from the underlying motivation for this work. Primarily, it seems the authors are highlighting that contamination detection methods are ineffective, yet it is unclear whether these methods are being used in practice. Recent work has highlighted pitfalls in data contamination approaches [1,2] and prohibiting training on test tasks [3]. A further discussion on how this work fits into prior work on fair evaluation and data contamination would strengthen the work.

In addition, do the authors believe that tailoring contamination detection methods will be the way forward for ensuring faithful evaluation when developers are incentivized to game benchmarks?

Minor
- Line 156 "Olypaid"

[1] Liu, Ken Ziyu, et al. "Language models may verbatim complete text they were not explicitly trained on." arXiv preprint arXiv:2503.17514 (2025).

[2] Fu, Yujuan, et al. "Does data contamination detection work (well) for llms? a survey and evaluation on detection assumptions." arXiv preprint arXiv:2410.18966 (2024).

[3] Dominguez-Olmedo, Ricardo, Florian E. Dorner, and Moritz Hardt. "Training on the test task confounds evaluation and emergence." arXiv preprint arXiv:2407.07890 (2024).

**Questions:**

- Could the authors clarify their claims in the Discussion and Conclusion? Specifically the claim "This fundamentally challenges the assumption that all the detection approaches rely on, which is that benchmark contamination is more about memorizing the benchmark samples."
- Could the authors provide additional details about why GRPO does not improve concealment on the Verbatim/Neighbor attacks?

---

> ### Author Response · Authors · 2025-11-25
> **Comments to Reviewer hTiN (1/N)**
>
> We thank the reviewer for their valuable suggestion. Below, we provide a detailed response regarding the motivation of our work and how our work is positioned in the literature.
>
> > **W1: Clarification of motivation**
>
> We clarify our motivations and contributions as follows.
>
> Recent work has proposed membership-inference attack methods as practical tools for checking whether evaluation data are used for training [a–g]. While several studies show that these methods are mostly ineffective for large-scale pre-training [a,b,c], potentially due to contaminated benchmark has been forgotten after training with trillions of tokens during the pre-training stage [i], **some other works demonstrate that they remain effective for detecting contamination in post-training [d, e, f, g]**, such as the supervised fine-tuning (SFT) pipeline. Recently, as Large Reasoning Models (LRMs) become popular and they are obtained precisely through post-training pipelines, typically involving SFT followed by reinforcement learning (RL), our work is **motivated by the research question: how reliable are the current contamination detection methods for LRMs, given that their training pipelines differ from prior settings by combining multiple post-training stages (SFT then RL) and extensive chain-of-thought style reasoning data?** We argue that this is exactly the regime where these detection methods are most likely to be deployed in practice.
>
> Within this context, our contributions are to **characterize the vulnerabilities of current detection methods in realistic LRM settings**. First, in the standard SFT then RL pipeline when the base model is evolving into LRMs, we show that contamination introduced during SFT is initially detectable (e.g., LiRA achieves ≈90% AUROC before GRPO. See Tables 2 and 7), but the subsequent RL can substantially conceal contamination signals. Second, for advanced LRMs, we find that strong benchmark gains can be achieved by performing SFT on CoT data while leaving almost no footprint for current detectors, despite significant exposure to test set data. Therefore, **our work complements previous works that show pitfalls of data contamination** methods such as [h] and reveals the vulnerability of current contamination detection methods from different perspectives.
>
> Finally, we believe there are many potential ways to ensure faithful evaluation, and we view tailored contamination detection methods as one of the important components of this toolkit, due to their ease of use and effectiveness [d, e, f, g]. Developing better contamination-detection methods can be used in an orthogonal manner to other approaches to ensure fair evaluation of LLMs.
>
> [a] Meeus, Matthieu, et al. "Sok: Membership inference attacks on llms are rushing nowhere (and how to treat fix it)." 2025 IEEE Conference on Secure and Trustworthy Machine Learning (SaTML). IEEE, 2025.
>
> [b] Das, Debeshee, Jie Zhang, and Florian Trantèr. "Blind baselines beat membership inference attacks for foundation models." 2025 IEEE Security and Privacy Workshops (SPW). IEEE, 2025.
>
> [c] Duan, Michael, et al. "Do membership inference attacks work on large language models?." arXiv preprint arXiv:2402.07841 (2024).
>
> [d] Puerto, Haritz, et al. "Scaling up membership inference: When and how attacks succeed on large language models." Findings of the Association for Computational Linguistics: NAACL 2025. 2025.
>
> [e] Mireshghallah, Fatemehsadat, et al. "An empirical analysis of memorization in fine-tuned autoregressive language models." Proceedings of the 2022 Conference on Empirical Methods in Natural Language Processing. 2022.
>
> [f] Fu, Wenjie, et al. "Membership inference attacks against fine-tuned large language models via self-prompt calibration." Advances in Neural Information Processing Systems 37 (2024): 134981-135010.
>
> [g] Panda, Ashwinee, et al. "Privacy auditing of large language models." arXiv preprint arXiv:2503.06808 (2025).
>
> [h] Dominguez-Olmedo, Ricardo, Florian E. Dorner, and Moritz Hardt. "Training on the test task confounds evaluation and emergence." arXiv preprint arXiv:2407.07890 (2024).
>
> [i] Bordt, Sebastian, et al. "How much can we forget about data contamination?." arXiv preprint arXiv:2410.03249 (2024).

---

> ### Author Response · Authors · 2025-11-25
> **Comments to Reviewer hTiN (2/N)**
>
> > **Q1: Clarifications of discussion and conclusion**
>
> We have revised our second claim in the Conclusion section to improve its clarity. Most detection methods rely on the assumption that contaminated models would achieve lower loss on training sequences [1] or generate less diverse responses for seen questions [2] than for unseen ones. Accordingly, these methods rely on a gap in certain metrics (e.g., log-probability, Levenshtein distance, etc.) between trained and unseen samples to determine contamination. However, benefiting from long CoT, we hypothesize that LRMs have strong generalization ability, making them easily generalize to unseen problems from distributions similar to the training data, rather than memorizing the fixed sequences. Thus, these LRMs could also have lower loss when responding to those unseen samples that share similar distributions to the training set, as shown in Figure 4. This confounding factor (i.e, generalization) is not accounted for by existing detection approaches, challenging the assumption that benchmark data contamination is more about memorization [3].
>
> [1] Carlini, Nicholas, et al. "Extracting training data from large language models." 30th USENIX security symposium (USENIX Security 21). 2021.
>
> [2] Dong, Yihong, et al. "Generalization or memorization: Data contamination and trustworthy evaluation for large language models." arXiv preprint arXiv:2402.15938 (2024).
>
> [3] Wu, Mingqi, et al. "Reasoning or memorization? unreliable results of reinforcement learning due to data contamination." arXiv preprint arXiv:2507.10532 (2025).
>
> > **Q2: Concealment of Verbatim/Neighborhood-based detection**
>
> For Verbatim/Neighbor attacks, the corresponding methods **already perform quite poorly and are essentially performing random guess with AUROC \approx 50% before GRPO**, shown in Table 2, indicating that they are not suitable to be deployed in the LRM contamination detection scenarios. As a result, GRPO has very little room to further degrade their performance.

---

> > ### Comment · Reviewer_hTiN · 2025-11-26
> >
> > Thanks for the response. While the result (pitfalls of data contamination methods) is not particularly surprising, the focus on LRMs in this work combined with the well-executed evaluation is satisfactory.

---

> > > ### Author Response · Authors · 2025-11-28
> > >
> > > Thanks. We are encouraged by your acknowledgment.

---

### Official Review · Reviewer_enZz · 2025-11-01

**Soundness:** 3
**Presentation:** 3
**Contribution:** 3
**Rating:** 6
**Confidence:** 3

**Summary:**

This paper investigates contamination detection methods for benchmarks in base models when they are being conditioned as LRMs via SFT and RL, and in advanced Large Reasoning Models via SFT with CoT contaminated data. They find that GRPO conceals contamination, and minimal detectable traces are left. 10 different detection methods and 6 mathematical reasoning benchmarks are used. A theoretical analysis is provided to show that PPO-style clipping and importance sampling are the root cause of the concealment. The paper makes the key claim that memorization based detection are not optimal for LRMs, and that these can evade it easily.

**Strengths:**

- It is highlighted that almost all the contamination detection methods (tested) consistently perform near random guesses in all the benchmarks after performing SFT on the benchmark samples with Co. This is a very significant insight.
- The paper is thorough in exploring its claim within the mathematical reasoning domain, using 6 common benchmarks in the field. And in using a wide range of detection methods (10).
- It is comprehensive in both providing results to support its claim and a thorough theoretical analysis of the premise

**Weaknesses:**

- The authors mention benchmark performance after SFT but do not mention results after SFT and GRPO.
- While enough benchmarks from the mathematical reasoning domain are used, no other domain is surveyed to make the points the paper raises applicable globally.

**Questions:**

- Do you have some figures on the length of outputs across the paper? Have you found any tested detection methods sensitive to output length?
- How do you generate your CoT RL data for each benchmark?
- In the first proposed direction in your conclusion (lines 477-478), how would the release of intermediate training checkpoints help with the issues the paper raises?
- Do you generate your own contaminated CoT and SFT data? If so, can you share it and how it was generated?
- Embedding based methods are mentioned but not used, why?

Suggestions:
- In line 72,can you clarify what you mean by “in the later stage”.
- In line 81, what kind of contamination is this? Is it verbatim contamination, paraphrased, etc..
- In the theoretical analysis it would be good to explicitly define E in formula 1 (and subsequent mentions).
- Given that this paper could aid malicious actors, can you provide more concrete recommendations for evaluators in the conclusion?

---

> ### Author Response · Authors · 2025-11-25
> **Comments to Reviewer enZz (1/N)**
>
> We thank the reviewer for their recognition of our work. We hereby added additional results on non-math domains and AUROC for embedding-based detections, and provide a point-to-point clarification on the reviewer's questions.
>
> > **W1: Benchmark performance after both SFT and RL**
>
> We report pass@1 under various settings in Table 1. In particular, every row with the RL data column that is not “/” corresponds to the performance after both SFT and GRPO. For example, row 2 denotes the pass@1 of SFT contamination first and then GRPO training with clean samples, and row 5 denotes the pass@1 of clean SFT training first and then GRPO with clean samples.
>
> > **W2: Applicability to non-math domains**
>
> We evaluate contamination effects beyond math and science domains. In addition to our main results on math benchmarks (Minerva Math, AIME24, AIME25, AMC23) and scientific benchmarks (OlympiadBench and GPQA-Diamond), we further examine **non-math domains at both pre-LRM and post-LRM stages**. We choose LiveCodeBench v2 (Coding) and MMLU-Pro (General QA). For MMLU-Pro, we only select the ‘health’, ‘history’, ‘law’, ‘other’, ‘philosophy’, and ‘psychology’ domains that are not related to math and science. The new experiments are **added to paper appendix F.9**, and also discussed below.
>
> First, we provide the pre-LRM stage (i.e., contamination happens when the base model evolves into LRMs) results and choose the Qwen2.5-7B-Instruct as the base model. Results shown in Table A indicate that the gap between member and non-member samples narrows with RL training, which means that **our claims extend to coding and general QA domains as well**. Overall, these results indicate that the **effect of RL concealments is not limited to the reasoning domains but extends to more general domains as well**.
>
> We also provide the AUROC of contamination detection approaches evaluated on advanced LRM that is contaminated in the post-LRM stage (i.e., contamination with CoT on advanced LRMs) in Table B. The results show that **contaminated LRMs leave minimal contamination evidence in coding as well**, indicating that the detection performance is near random guess for those reasoning-related domains (e.g., coding, math, science, etc). **For domains that require less CoT (e.g., general QA)**, despite that the AUROC could reach 75% when using the Loss as the detection for the contaminated Deepseek distilled Qwen-14B, **the detection performance is much worse compared to the results in the pre-LRM stage**. Specifically, the AUROC for the Loss detector on MMLU-Pro can reach 93.72% when SFT contamination is applied to the base model, shown in Table A. We believe these results further validate our point that contaminated LRMs leave less detectable evidence of contamination compared to SFT contamination on the base model due to LRMs’ ability to generate long CoT and generalization capabilities.

---

> ### Author Response · Authors · 2025-11-25
> **Comments to Reviewer enZz (2/N)**
>
> Table A: AUROC (%) of contamination detection approaches evaluated starting from an SFT-contaminated model w/o RL to subsequently trained with GRPO **in pre-LRM stage**. Results demonstrate that after GRPO, AUROC decreases across **both general QA and coding domains** and different detection approaches. Δ measures the difference with the SFT-contaminated model w/o RL. Higher AUROC, better detection performance. The base model here is Qwen2.5-7B-Instruct.
>
> | Method | Training Stage | MMLU-Pro (Non-STEM domains) | LiveCodeBench | Avg.   | Δ      |
> |--------|----------------|-----------------------------|---------------|--------|--------|
> | LiRA   | Before RL      | 96.68                       | 79.85         | 88.265 | 0      |
> | LiRA       | RL w/ clean    | 93.85                       | 71.12         | 82.485 | -5.78  |
> | Ref    | Before RL      | 86.25                       | 69.51         | 77.88  | 0      |
> | Ref       | RL w/ clean    | 81.67                       | 66.68         | 74.175 | -3.705 |
> | Loss   | Before RL      | 93.72                       | 64.87         | 79.295 | 0      |
> |  Loss      | RL w/ clean    | 88.69                       | 54.26         | 71.475 | -7.82  |
> | Max-K  | Before RL      | 82.27                       | 63.75         | 73.01  | 0      |
> |  Max-K      | RL w/ clean    | 59.48                       | 49.8          | 54.64  | -18.37 |
> | Min-K  | Before RL      | 93.43                       | 64.12         | 78.775 | 0      |
> |  Min-K      | RL w/ clean    | 88.54                       | 54.29         | 71.415 | -7.36  |
>
> Table B: AUROC (%) of contamination detection approaches evaluated on advanced LRMs contaminated with CoT in both **general QA and coding domain** benchmarks **in post-LRM stage**.
>
> | Method | Init Models  | MMLU-Pro (Non-STEM domains) | LiveCodeBench | Avg.  |
> |--------|--------------|-----------------------------|---------------|-------|
> | LiRA   | DS Llama-8B  | 72.85                       | 61.37         | 67.11 |
> | LiRA   | DS Qwen-14B  | 74.75                       | 58.78         | 66.77 |
> | Ref    | DS Llama-8B  | 60.77                       | 58.75         | 59.76 |
> | Ref   | DS Qwen-14B  | 63.43                       | 58.17         | 60.80 |
> | Loss   | DS Llama-8B  | 69.56                       | 57.42         | 63.49 |
> | Loss   | DS Qwen-14B  | 75.83                       | 56.52         | 66.18 |
> | Min-K  | DS Llama-8B  | 68.12                       | 56.59         | 62.36 |
> | Min-K  | DS Qwen-14B  | 76.09                       | 56.08         | 66.09 |
> | Max-K  | DS Llama-8B  | 61.07                       | 56.33         | 58.70 |
> |  Max-K | DS Qwen-14B  | 66.88                       | 54.80         | 60.84 |

---

> ### Author Response · Authors · 2025-11-25
> **Comments to Reviewer enZz (3/N)**
>
> > **Q1: Output length**
>
> We **add a new visualization of the average response length per question** across different benchmarks in Figure 6, Appendix F.3. Although the **average response length does not change too much after GRPO training**, several detection approaches already exhibit a substantial performance drop, as shown in Table 2. In contrast, for RAFT,  the response lengths remain similar to those after GRPO but no concealment effect is observed. These results suggest that response **length is not the key factor**; instead, the entropy of the model’s output appears to drive the concealment phenomenon.
>
> >**Q2: RL CoT data**
>
> During the RL phase, **we follow a typical GRPO pipeline, and no external CoT data was added. The CoT is generated by the model itself during the rollout phase in RL training**. We define RL contamination as the model has encountered the benchmark question and has received rewards based on its generated responses and the groundtruth answer during RL finetuning (beginning of Section 3).
>
> > **Q3: Releasing intermidiate checkpoints**
>
> Current reasoning models typically undergo both SFT and RL stages to acquire strong reasoning ability [1]. If model developers only release checkpoints after RL, it becomes very hard for the community to determine whether the model has contaminated the benchmark or not, due to the contamination-concealing effect of RL training. If model developers instead release more intermediate checkpoints, e.g., checkpoints after cold-start SFT (before RL), the community may use these earlier checkpoints to detect potential contamination more reliably. However, this approach still ultimately relies on developer integrity, as they could release a clean intermediate checkpoint while introducing contamination in later stages.
>
>
> > **Q4: CoT and SFT data**
>
> We use the QwQ/Qwen-32B as the oracle reasoning models to help generate the responses to the benchmarks. We adopt the temperature 0.6, top-p value of 0.95, and maximum output token with 32768. We use 64 rollouts for each question, selecting correct trajectories when available. If none of the 64 rollouts produce a correct answer, we randomly select an incorrect trajectory. More details could be found in Appendix E.1 and Table 6.
>
> > **Q5: Embedding-based detection**
>
> In our original submission, we **provided the UMAP visualization of the hidden state of the last token of question+response in the last layer in Appendix F.8**. Here, we provide more quantitative results to evaluate the effectiveness of the embedding-based methods [2] in the pre-LRM stage with Qwen2.5-7B-Instruct. We randomly split all member and non-member examples into training and test sets, maintaining a consistent 0.8/0.2 ratio across datasets, and train an MLP classifier to predict the member/non-member labels. The input to the MLP is the hidden state feature of the last token in the question and response pair. Following [2], we evaluated several hidden layers and selected the best-performing one. We report results only on the medium-scale benchmarks (Olympiad, GPQA, Minerva), since the small-scale datasets have very few test examples, resulting in too much variance. **The results show that contamination is initially detectable, but becomes undetectable after RL, which is consistent with our paper**. We have added the results in Appendix F.8.
>
> Table C: AUROC (%) of the **embedding-based detection** [2] evaluated starting from an SFT contaminated model w/o RL to subsequently trained with GRPO in pre-LRM stage. Results demonstrate that after GRPO, AUROC decreases. Δ measures the difference with the SFT-contaminated model w/o RL. Here, the base model is Qwen2.5-7B-Instruct.
>
> | Training Stage      | Olympiad | GPQA   | Minerva | Avg.    | Δ       |
> |---------------------|----------|--------|---------|---------|---------|
> | Before RL           | 58.89   | 80.00    | 60.05  | 66.31  | +0.00   |
> | RL w/               | 56.63   | 45.75 | 49.60  | 50.66  | -15.65  |
> | RL w/ Clean&Mem     | 57.43  | 45.00    | 48.94  | 50.46  | -15.85  |

---

> ### Author Response · Authors · 2025-11-25
> **Comments to reviewer enZz (4/N)**
>
> > **Suggestion 1: Clarification on "later stage"**
>
> Since usually there are two training stages for the reasoning models [1], the earlier stage can be the SFT, and the later stage can be the RL. We have modified this sentence to make the stage clearer.
>
> > **Suggestion 2: Clarification on definition of contamination**
>
> Contamination in line 81 refers to the SFT and RL contamination. We define the SFT and RL contamination for the reasoning model at the beginning of Section 3: **For the SFT contamination, we define it as the model being exposed to both the benchmark question and responses distilled from an advanced LRM, where RL contamination refers to the model encountering the benchmark question and having received rewards based on its generated responses during RL finetuning.**
>
> Here, it is neither verbatim contamination nor paraphrased contamination. Since many reasoning benchmarks only release the question and ground truth answers (e.g., AIME2025), while LRMs rely on the CoT to reach the final answer, we need to define the benchmark contamination on the LRMs differently from previous verbatim (i.e., training on exact solution released by the benchmark developers) or paraphrase contamination (i.e., training on paraphrased solution).
>
> > **Suggestion 3: Definition of E in formula 1**
>
> Thanks for pointing out the unclarity. The $\mathbb{E}$ in Equations (1), (2), (3), and (4) indicates the expectation operator. Intuitively, formula (1) expresses the average negative log likelihood difference between the answers to the non-member data and member data, conditioned on the answer being correct.
>
> > **Suggestion 4: Recommendation for evaluators**
>
> Given the insights shown in our paper, we believe that the research community should develop advanced contamination detection approaches for LRMs that explicitly account for the long CoT reasoning and generalization capability. Beyond this, we see at least three potential directions for model evaluators: (1) most importantly, model evaluators should be aware that existing contamination detection methods may not be effective for LRMs. (2) before better contamination detection methods are developed, evaluators could choose not to publicly release the benchmarks and instead evaluate LRMs on private benchmarks. (3) developing dynamic benchmarks which do not have a fixed set of questions [3,4] can be a potential solution to better evaluate the capabilities of LRMs.
>
> ### Reference
> [1] Guo, Daya, et al. "Deepseek-r1: Incentivizing reasoning capability in llms via reinforcement learning." arXiv preprint arXiv:2501.12948 (2025).
>
> [2] Liu, Zhenhua, et al. "Probing language models for pre-training data detection." Proceedings of the 62nd Annual Meeting of the Association for Computational Linguistics (Volume 1: Long Papers). 2024.
>
> [3] Huang, Kaixuan, et al. "MATH-Perturb: Benchmarking LLMs' Math Reasoning Abilities against Hard Perturbations." arXiv preprint arXiv:2502.06453 (2025).
>
> [4] Zou, Chengke, et al. "Dynamath: A dynamic visual benchmark for evaluating mathematical reasoning robustness of vision language models." arXiv preprint arXiv:2411.00836 (2024).

---

### Author Response · Authors · 2025-11-25

**Overall comment:**

We thank all reviewers for their constructive feedback. During the rebuttal, we have **tried our best to provide new experiments** on the applicability in non-math domains, provide results for extensive RL training, and analyze contraction (i.e., the log-probability gap between member/non-member shrinks) for smaller/larger models as reviewers request:

1. **Applicability in non-math domains (Review enZz, F3s6):** We provide results on coding and general QA domains and find that our conclusion in both pre-LRM and post-LRM extends beyond math and science domains as well. (Paper Appendix F.9)

2. **Concern about RL training (review F3s6):** We provide results that more RL steps training could bring in performance gain. (Paper Appendix F.11)

3. **Contraction among large/small models (Review F3s6):** We provide results when the SFT contamination on base models with 3B and 14B in the pre-LRM stage and find that our conclusion extends to a large spectrum of model sizes. (Paper Appendix F.10)

---

### Author Response · Authors · 2025-12-04
**Rebuttal Summary for the new AC**

We thank all the reviewers for their careful reading and constructive comments, and we thank the AC for their additional effort under this unusual situation. Below, we briefly summarize how we addressed the main concerns of each reviewer.

---

### **Reviewer enZz (Score 6 -> unknown)**

**Main concerns:**

1. Some details about the pass@1 after RL, SFT/RL contamination, and why releasing intermediate checkpoints helps
2. Question about the applicability of the conclusion in non-math domains.
3. Length of the output responses.
4. AUROC of embedding-based methods.

**Our response:**

We addressed these points by:
1. Directing the reviewer to the specific session in the paper for more details
2. **Providing more results to show that our conclusions in both pre-LRM and post-LRM stages are still valid in non-math domains**
3. Providing visualization of the output length to show that the response length is not the key factor; instead, the entropy of the model’s output appears to drive the concealment phenomenon.
4. Providing quantitative results for an embedding-based detection, and we observe that it initially detects contamination quite well, but becomes undetectable after RL, which is consistent with our paper
---

### **Reviewer hTiN (Score 6 -> 8)**

**Main concerns:**

1. Question about the motivation, and why the assumption of existing detection approaches does not work in the reasoning model scenarios.

**Our response:**

1. We provide more clarification of the motivation, discussion, and conclusion in the rebuttal.
---

### **Reviewer F3s6 (Score 8 -> unknown)**

**Main Concerns:**

1. Why does RL not bring significant performance gain in Table 1
2. The applicability of conclusions in the non-math domains
3. The contraction (i.e., the log-probability gap between member/non-member shrinks) on larger/smaller models

**Our response:**
1. We clarify that the insignificant RL improvement was due to our relatively short RL training run to save experiment cost, which already effectively demonstrated the contamination concealment effect. And we provide more experiments to further verify our RL pipeline by continuing RL training on the SFT-contaminated Qwen-3B model for up to 280 steps, and show that **RL continues to improve task performance as well as conceal the contamination**.
2. We **add more results to show that our conclusion in post-LRM stages extends beyond the math domain**
3. We **add more results to further validate that the contraction extends to a larger spectrum of model sizes.**

---

### Meta-Review · Area_Chair_ukFU · 2025-12-18

**Summary:**

The reviewers agree the paper addresses the important problem of benchmark contamination with well-designed experiments. There are some concerns on the motivation, technical details, and benchmarking beyond mathematical reasoning, but the authors thoroughly address them with new experiments. All reviewers are satisfied with the paper as evidenced by the high scores.

**Reviewer Concerns:**

The concerns on motivation, technical details, and benchmarking beyond mathematical reasoning have been sufficiently addressed with new experiments. The AC does not see any concerns that are still outstanding.

**Reviewer Scores:**

* Reviewer enZz: the score is 6
* Reviewer hTiN: based on the discussion, it is clear the score increased from 6 to 8
* Reviewer F3s6: the score is 8

---

### Decision · Program_Chairs · 2026-01-26

Accept (Poster)